# QueST: Self-Supervised Skill Abstractions for Learning Continuous Control

**Atharva Mete[1], Haotian Xue[1], Albert Wilcox[1], Yongxin Chen[1,2], Animesh Garg[1,2]**

[1]Georgia Institute of Technology, [2]NVIDIA

## Abstract

Generalization capabilities, or rather a lack thereof, is one of the most important unsolved problems in the field of robot learning, and while several large scale efforts have set out to tackle this problem, unsolved it remains. In this paper, we hypothesize that learning temporal action abstractions using latent variable models (LVMs), which learn to map data to a compressed latent space and back, is a promising direction towards low-level skills that can readily be used for new tasks. Although several works have attempted to show this, they have generally been limited by architectures that do not faithfully capture sharable representations. To address this we present Quantized Skill Transformer (QueST), which learns a larger and more flexible latent encoding that is more capable of modeling the breadth of low-level skills necessary for a variety of tasks. To make use of this extra flexibility, QueST imparts causal inductive bias from the action sequence data into the latent space, leading to more semantically useful and transferable representations. We compare to state-of-the-art imitation learning and LVM baselines and see that QueST's architecture leads to strong performance on several multitask and few-shot learning benchmarks. Further results and videos are available at https://quest-model.github.io.

## 1 Introduction

One of the grand goals of robotic learning is a general-purpose model that can learn from complex multitask demonstration data and generalize to new tasks in a zero-shot or few-shot manner. While such general-purpose models have become ubiquitous in natural language (NLP) [66, 72, 52, 53, 63] and computer vision (CV) [77, 33, 7], they have eluded robotics researchers. Whereas CV and NLP can achieve positive transfer by scaling up models trained on internet scale datasets [33, 72, 56, 62, 39], even large scale robot data collection efforts [10, 11, 47, 20, 31] have been insufficient for this approach. To that end, we posit that in order to achieve positive transfer in robotics, it is important to design architectures that specifically lend themselves to efficient cross-task transfer.

There has recently been a surge of work towards the goal of learning generalist policies from large, diverse datasets. Several papers have used techniques such as action discretization [57, 16, 35, 10, 11, 47, 55] and implicit models [23, 15, 25] to model multimodal action distributions. In particular, the behavior transformer line of work shows that a carefully discretized action space combined with a GPT-style transformer leads to impressive capabilities modeling multimodal behavior distributions [57, 16, 35]. In another vein, several works have attempted to scale demonstration data and achieve positive transfer of low-level skills between high-level tasks [10, 11, 47, 46, 18, 12, 55, 3]. While these works have shown some transfer, for example applying policies for known tasks to unfamiliar objects, they have generally failed to achieve transfer of low-level skills to novel tasks [4]. We

---

[1]Correspondence to: amete7@gatech.edu

38th Conference on Neural Information Processing Systems (NeurIPS 2024).

hypothesize that in the relatively low-data regime of robot learning, it is promising to explicitly force the model to learn sharable representations. To that end we study latent variable models (LVMs), which learn to map data to a compressed latent space and back, introducing an information bottleneck which encourages the model to learn shared representations across the training data. Specifically we consider the application of LVMs to learn low-dimensional representations of action sequences. Such representations are termed temporal action abstractions or motion primitives – in this paper we refer to these abstractions as 'skills'.

A wide body of work has considered the application of LVMs to robotics. One line of work learns temporal action abstractions (skills) in continuous latent space with a Gaussian prior [41, 50, 59]. While this line of work showed some initial promise of learning latent plans, it has failed to scale to difficult multitask settings due to the loose nature of the latent structure and posterior collapse issues that inhibit the learning of shared representations. On the other hand, recent work in CV and NLP has shown that vector-quantized discrete latent spaces are capable of learning semantically meaningful representations from data like phonetics in speech [9, 6] or melody in music [17, 2]. This insight, along with prior work showing that discretized action spaces can help to address the multimodality problem in when learning from large datasets [57, 16, 35, 14], motivates methods learning temporal action abstractions with discrete latent spaces. Several recent works have set out to do this [35, 74, 76, 30], showing some degree of positive transfer between tasks in multi-task and few-shot settings. However, they are generally limited by architectures that do not faithfully capture transferable representations [35, 74], or depend on state prediction and state-based objective functions which are impractical for many real robot tasks [76, 30].

In this paper, we present **Qu**antized **S**kill **T**ransformer (QueST), a simple yet novel architecture for learning generalizable low-level skills within a discrete latent space. The key insight behind QueST is its ability to flexibly capture variable length motion primitives by representing them with a sequence of discrete codebook entries. We achieve this through a unique encoder-decoder architecture primarily designed to impart causal inductive bias in action sequence data into the latent space. Such formulation enables us to employ powerful sequence modeling approaches to plan and composably reason within the space of low-level skills. Through our experiments, we show that autoregressive modeling of these latent skills with a GPT-like transformer outperforms state-of-the-art baselines on challenging robotic manipulation benchmarks, where QueST shows an 8% improvement in multitask and 14% improvement in few-shot imitation learning over the next best baselines. We also conduct a detailed ablation and sensitivity study to validate our key architectural design decisions.

## 2 Related Works

The proposed framework in this paper introduces a methodology for self-supervised skill abstraction, followed by decision-making within this skill space. Several related works have explored similar sub-directions such as decision-making in the latent space and decision making with a transformer:

### 2.1 Learning from Offline Data

Behavior cloning (BC) [51] aims to learn a policy by directly mapping observations to actions, and is typically trained end-to-end using pre-collected pairs of observation and behavior data. While this on its surface is a simple supervised learning problem, there are several properties of robot demonstration data that should be considered when building BC systems. First, large BC datasets collected from a variety of human demonstrators tend to contain data sampled from multimodal distributions. To address this, some works opt to sample actions from Gaussian Mixture Models (GMM) [43], while others explore implicit models including those derived from energy-based models [23, 29] or diffusion models [15, 71, 14, 68, 28]. The Behavior Transformer (BeT) line of work [57, 16, 35] shows that transformer-based categorical policies in carefully discretized action spaces do a good job handling multimodal demonstrator distributions and QueST builds upon this by contributing a more capable discrete latent skill model.

Another key property of robot demonstration data is that sequential actions are often highly correlated with one another, and exploiting this can lead to stronger performance while ignoring it can lead to policies which are susceptible to temporally correlated confounders [61]. Recently several works have set out to handle this by predicting action chunks. For example, the Action Chunking Transformer (ACT) line of work [73, 24] shows that a transformer trained as a CVAE [60] to output chunks of

actions performs well for a wide variety of manipulation tasks, and diffusion policy [15] shows across the board improvements when predicting action chunks. As discussed in detail in Section 2.3, QueST builds on a long line of work which handles sequential correlations through temporally-extended action abstractions [41, 50, 59, 35, 74, 76, 30].

## 2.2 Multi-task and Few-shot Imitation Learning

In the past, robot learning researchers have approached multi-task decision making settings using a wide variety of methods such as supervised pre-training and fine-tuning [20, 42], meta-learning [22, 19] and action retrieval [48, 44]. There has recently been a large focus on multi-task language-conditioned imitation learning for robotics with several papers attempting to address the problem by training large models on large demonstration datasets [10, 11, 47, 46, 18, 12, 1]. While these papers achieve impressive multitask results, they mostly rely on sufficient data coverage and fail to generalize beyond their training distribution [4]. Thus, they lack abstractions that can readily be applied to learn new tasks, especially in a low-data regime. On the other hand LVMs like QueST are designed to learn sharable representations that can be applied to new tasks.

## 2.3 Decision Making in Learned Latent Spaces

LVMs, modeled by a paired encoder-decoder, have found extensive applications in computer vision [65, 32, 8] and generative models [56, 21, 13]. Recent studies also demonstrate the utility of latent space representations in robot decision-making, spanning offline RL [58, 50, 40], imitation learning [14, 67, 37, 35], and temporal action abstraction [57, 74, 75]. Most similar to our work are those that learn temporally abstracted discrete latent skill spaces. PRISE [74] learns single-step state-action abstractions within a discrete space and then does temporal abstraction by applying BPE tokenization. While this method shows promise in learning multi-task and few-shot policies, BPE is well suited for text and is known to suffer in domains with highly dynamic vocabularies, in robotics its equivalent to varying action distributions across unseen tasks. TAP [76] and H-GAP [30] utilize a self-supervised auto-encoder to learn skill codes, but their functionality relies on Model Predictive Control (MPC) using state prediction with ground-truth state-conditioned objective functions, which make it difficult to apply to real-world manipulation tasks. VQ-BeT [35] bears the strongest resemblance to QueST, also using a pre-trained discrete latent skill space to discretize the action space for a transformer-based policy prior. However, their quantization approach does not leverage the inherent structure in action sequences, limiting the representational capabilities of the latent space. Thus it's shown to heavily rely on an continuous offset predictor for best performance. Unlike these works, QueST's learned latent space is highly flexible yet structured and expressive, allowing it to effectively model action distribution across many distinct tasks in a meaningful shared representation.

# 3 Preliminary

## 3.1 Problem Setting

We consider a dataset $D = \{\{(O_1, a_1), \ldots, (O_{T_i}, a_{T_i})\}_{i=0}^{N_k}, L_k\}_{k=0}^{M}$ consisting of $M$ robot interaction trajectories where $a_t$ is a continuous-valued action and $O_t$ is a tuple consisting of a high-dimensional sensory observation. The data is collected via either human teleoperation or scripted policies for M different task each with a label $L_k$. In our setting, $O_t$ consist of RGB image observations from the front camera and gripper camera (if available) along with proprioceptive state of the agent. $L_k$ is a natural language description of the $k^{th}$ task but can also be a one-hot encoding.

## 3.2 Finite Scalar Quantization

We build on Finite Scalar Quantization (FSQ) [45] as a discrete bottleneck in our model. It's a drop-in replacement for Vector Quantization (VQ) layers in VAEs with a simple scalar quantization scheme. Here the input representation $e$ is projected to very few dimensions (typically 3 to 5) which are then bounded and rounded, creating an implicit codebook.

$$z = \text{round\_ste}(f(e)), \text{ where } f \text{ is the bounding function} \tag{1}$$

Given a feature vector $e \in \mathbb{R}^d$ to quantize, instead of learning a parameterized codebook [65] and quantizing $e$ by matching the nearest neighbor in the codebook, FSQ quantizes $e$ by first bounding

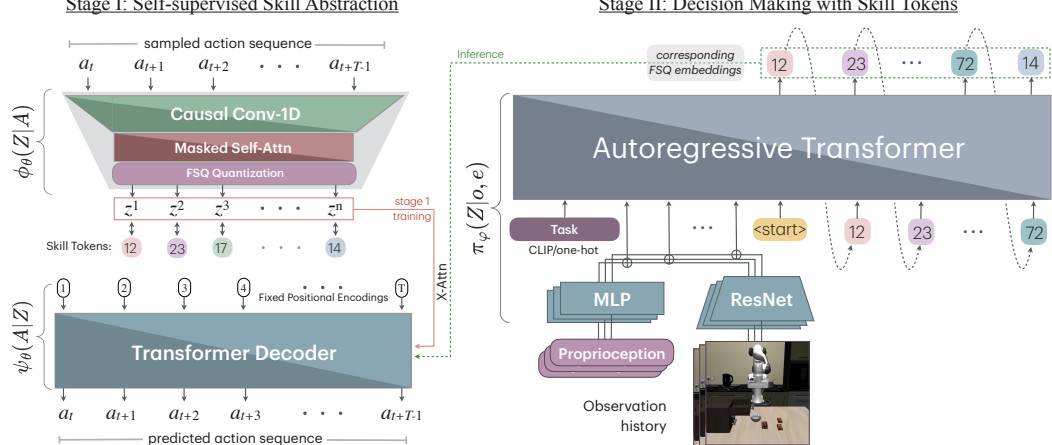

Figure 1: **Overview of Quantized Skill Transformer**: we factorize the policy that outputs action based on task descriptions $e$ and observations $o$ encoding into two parts: $\pi(A|o,e) = \psi_\theta(A|Z)\pi_\varphi(Z|o,e)$, where $Z$ is a sequence of skill tokens for the action sequence $A$. In Stage I, we learn skill abstraction in a self-supervised way with a quantized autoencoder. In Stage II, we learn skill-based policy in the style of next-token prediction using a multi-modal transformer.

it into certain range with $f$ (e.g. $f = \lfloor \alpha/2 \rfloor \odot \tanh(e)$) and then rounding each dimension into integer numbers directly with straight-through gradients (round_ste). $\alpha \in \mathbb{Z}^d$ defines the width of the codebook for each dimension (e.g. $\alpha = [8, 5, 5, 5], d = 4$). Finally, it is easy to see that the size of the quantization space is $\Pi_{i=1}^d \alpha_i$. An MLP can be used to transform $z$ into continuous space of required dimension further.

A common problem with vector-quantized codebooks (VQ) [65] is the under-utilization of the codebook. Recent works have attempted to address by heuristics like reinitializing the codebook, stochastic formulations, or some regularization [34, 69]. In contrast, FSQ achieves much better codebook utilization for large codebook sizes with much fewer parameters and simplified training without any auxiliary losses or aforementioned tricks. Due to its simplicity and proven benefits, we use FSQ in our main experiments, but since many prior works in this space use VQ we also perform an ablation with it (see section 5.6).

## 4  Method

In this section, we describe the key ideas behind Quantized Skill Transformer. In Section 4.1 we present our encoder-decoder architecture, which is designed to provide the flexibility to learn a wide range of skills with inductive biases to ensure that the learned skills are useful. In Section 4.2, we detail our skill prior, which we train to autoregressively predict codebook skills. Our full pipeline is shown in Figure 1.

### 4.1  Stage I: Learning the Skill Codebook

As a motivating example, consider the task of lifting a pot and placing it on a stove beside. This consist of primitives like reaching the pot, grasping it, lifting it to a certain height, reaching the stove and finally placing it on the stove. Each of these primitives are of variable lengths, and to properly model these skills it is important to learn a latent skill space with the flexibility to model all of them. At the same time, it is important that the learned skills are semantically meaningful so that they can be reused for new tasks, for example reusing the reaching skill for an object lifting task. In order to address these desiderata, we introduce the novel autoencoder architecture shown in Figure 1 consisting of an encoder $\phi_\theta$ and decoder $\psi_\theta$.

The input to the encoder $\phi_\theta$ is an action sequence $a_{t:t+T-1}$ sampled from the dataset, which we pass through several 1D causal strided-convolution layers [64]. This step reduces the sequence length to achieve the desired temporal abstraction depending on the stride lengths and the number of layers.

We follow the convolutional layers with masked self-attention layers for sequence modeling. With a downsampling factor of $F$, the encoder outputs in total $n = T/F$ embeddings. The embeddings are then quantized using FSQ as per the equation 1 into $n$ discrete latent codes $\{z^i\}$ termed as skill tokens:

$$(z^1, \ldots, z^n) = \text{FSQ}(\phi_\theta(a_t, \ldots, a_{t+T-1})). \tag{2}$$

Having an input sequence of actions mapped to multiple skill tokens gives this architecture more flexibility to model complex sequences of actions. At the same time, each component of the encoder is causal, meaning that an output representation at a position $t$ cannot depend on input from any future timesteps. We found this inductive bias to encourage the model to learn semantically useful action representations by modeling the inherent causality in the action data. We validate this design choice in the ablations. (see section 5.6)

Typical autoencoder decoders are simply mirrored versions of the encoders, but this would prevent the decoder from attending to all quantized codes. This is important because individual codes do not represent anything meaningful but a sequence of codes represents a particular meaningful motion [45]. In order to maintain causality while attending to all codes, the decoder $\psi_\theta$ cross attends between fixed sinusoidal positional embedding inputs and the skill tokens, similarly with [73]. The architecture is a transformer decoder block consisting of alternate masked self-attention and cross-attention layers, after which the output embeddings are projected back to the original action dimension using an MLP layer. Thus, given a sequence $Z$ of skill codes, $\psi_\theta$ reconstructs the original action

$$(\hat{a}_t, \ldots, \hat{a}_{t+T-1}) = \psi_\theta(z^1, \ldots, z^n) \tag{3}$$

As in [35], the autoencoder is trained by minimizing the $\ell_1$ reconstruction loss:

$$\mathcal{L}_{\text{recon}}(\theta) = \|\psi_\theta(\text{FSQ}(\phi_\theta(a_{t:t+T-1}))) - a_{t:t+T-1}\|_1. \tag{4}$$

Unlike prior work which often conditions on the state as well as the actions [30, 5, 74, 37], we choose to learn state-independent abstractions that solely capture motion primitives irrespective of the current scene or task. Through our experiments we show that our model learns generalizable abstractions that are shared and can be transferred across tasks.

## 4.2   Stage II: Learning the Skill Prior

After training the encoder $\phi_\theta$ and decoder $\psi_\theta$, we train a skill prior $\pi_\varphi(Z|e, o)$ to predict skills $Z = z^{1:n}$ corresponding to the demonstrator action distribution conditioned on a task embedding $e$ and a length $h$ sequence of image observations and proprioception inputs, $o = (i_{t-h}, p_{t-h}), \ldots, (i_t, p_t)$. We encode image observations with a separate learned vision encoder for each camera view and encode proprioception using an MLP encoder, all of which are trained end-to-end with the rest of the skill prior. The observation token $\mathcal{T}_t^o$ for a timestep $t$ is obtained by concatenating outputs from all the aforementioned encoders. Task embeddings are designed specifically for each environment suite, as discussed in more detail in Section 5. See Appendix B for more details about the encoders.

Because skill tokens are highly dependent on one another according to the complex nonlinear representations learned by the autoencoder, it is important that the skill prior has the modeling capacity to reason about these dependencies. To achieve this, we employ a decoder-only transformer to model the distribution of skill tokens $\pi_\varphi(Z|\mathcal{T}_{t-h:t}^o, e)$ autoregressively as:

$$\pi_\varphi(Z|\mathcal{T}_{t-h:t}^o, e) = \prod_{i=1}^{n} \pi_\varphi(z^i|\texttt{}, z^{1:i-1}, \mathcal{T}_{t-h:t}^o, e) \tag{5}$$

where $\texttt{}$ is a learnable start token that marks the start of skill tokens. We add sinusoidal positional embeddings only to the skill tokens. To optimize the skill prior, we sample a sequence of demonstrator actions $a_{t:t+T-1}$ and use the trained encoder $\phi_\theta$ to extract a latent skill vector $Z_t = z^{1:n}$ according to Equation 2. Then, we optimize $\pi_\varphi$ using the following negative log-likelihood loss:

$$\mathcal{L}_{\text{task}}(\varphi) = -\log \pi_\varphi(Z_t|\mathcal{T}_{t-h:t}^o, e). \tag{6}$$

The full skill prior pipeline is shown in Figure 1.

**Few-Shot Finetuning:** For few-shot finetuning on new tasks, we use a model pre-trained on large set of tasks and finetune it on a small number of demonstrations (5 in our experiments) from the held-out task. Although finetuning only stage-2 is enough, we empirically found that finetuning the decoder on the predicted skill tokens gives a boost in the performance. Specifically, we finetune the decoder using following decoder loss:

$$\mathcal{L}_{\text{decoder}}(\theta) = \|\psi_\theta(\text{sg}(\hat{Z}_t)) - a_{t:t+T-1}\|_1 \tag{7}$$

where sg is the stop gradient operator. We present the results with and without decoder finetuning both. Additionally, we note that the encoder is still frozen in this setting.

### 4.3 Inference with Quantized Skill Transformer

At inference time, QueST uses the skill prior $\pi_\varphi$ alongside the decoder $\psi_\theta$ to sample actions. Conditioned on the encoded observation history $\mathcal{T}^o_{t-h:t}$ and task embedding $e$, we use top-$k$ sampling with a temperature of $\tau$ to autoregressively sample a skill vector $\hat{Z} \sim \pi_\varphi(\cdot|\mathcal{T}^o_{t-h:t}, e)$ from the skill prior. In practice, we find $k = 5$ and $\tau = 1$ to work well across all environments. Then, we use the decoder to map the skill vector back to the action space, producing a sequence of predicted actions $\hat{a}_{t:t+T-1} = \psi_\theta(\hat{Z})$. In a receding horizon fashion, we execute the first $T_a \leq T$ actions before replanning.

## 5 Experiments

We design the experiments to empirically evaluate the performance of Quantized Skill Transformer in three practical settings: (1) Multitask IL, (2) Few-shot transfer, and (3) Long-horizon IL. Lastly, we perform some ablations to empirically justify our model design choices.

### 5.1 Benchmarks and Baselines

We use the following benchmark suites to evaluate in the settings discussed above:

**LIBERO [38]** is a lifelong learning benchmark featuring several task suites consisting of a variety of language-labeled rigid- and articulated-body manipulation tasks. Specifically, we evaluate on the LIBERO-90 suite, which consists of 90 manipulation tasks, and the LIBERO-LONG suite, which consists of 10 long-horizon tasks composed of two tasks from the LIBERO-90 suite. As described in more detail below, we use the LIBERO benchmark to study the multitask IL, few-shot transfer and long-horizon IL settings. Because tasks from this benchmark are language-annotated, we use the output of a frozen CLIP [54] encoder for the task conditioning input $e$.

**MetaWorld [70]** features a wide range of manipulation tasks designed to test few-shot learning algorithms. We use the Meta-Learning 45 (ML45) suite which consists of 45 training tasks and 5 difficult held-out tasks which are structurally similar to the training tasks. We use this benchmark to test multi-task and few-shot learning. Because this benchmark does not include language labels, we use learned task embeddings $e$ for task conditioning.

**Baselines:** We compare to the following baselines, which include similar discrete LVM pipelines as well as state of the art imitation learning algorithms:

1. The **ResNet-T** model from [38], which encodes observation and task instructions using ResNet-18 with FiLM [49], applies a transformer sequence model, and uses a GMM head to predict actions.

2. The UNet-based **diffusion policy** from [15], which uses a 1D convolutional UNet to map samples from a Gaussian prior to action samples from the demonstrator distribution according to a learned denoising process.

3. **ACT [73]**, which trains a transformer as a CVAE [60] to predict action chunks.

4. **VQ-BeT [35]**, which learns a discrete latent space using a VQ-VAE [65] and uses a transformer to predict discrete latent codes.

5. **PRISE [74]**, which first quantizes observation-action pairs and performs temporal abstraction using byte pair encoding (BPE) to learn a skill token vocabulary, which it uses as an action space for few-shot learning.

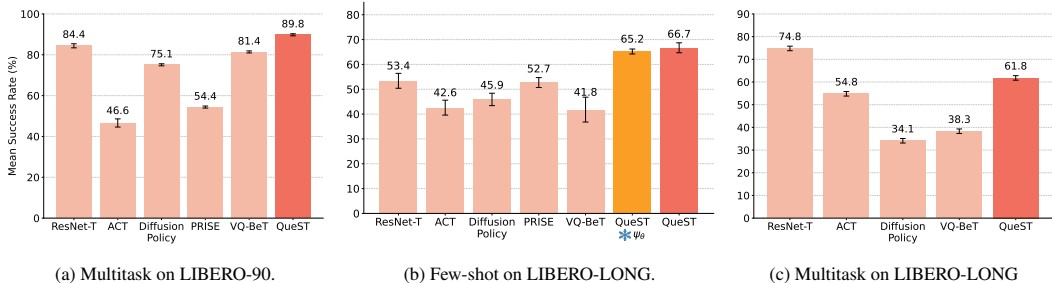

(a) Multitask on LIBERO-90.     (b) Few-shot on LIBERO-LONG.     (c) Multitask on LIBERO-LONG

Figure 2: Multitask performance on LIBERO-90 (a) and LIBERO-LONG (c), and few shot performance on LIBERO-LONG (b). For (a) and (c) we train on the datasets described in Sections 5.2 and 5.4. For (b) we finetune the model from (a) on a condensed dataset as described in Section 5.3. Results show the mean and error bar represents standard error across four random seeds for multitask experiments and nine random seeds for fewshot experiments. Results for PRISE are taken from Zheng et al. [74], and the others we reimplemented and ran ourselves.

For a detailed discussion between QueST and the baseline methods, please refer to Appendix B.5.

## 5.2 Performance on Multitask BC

We evaluate the goal-conditioned multi-task imitation learning capabilities of QueST and the baselines using the LIBERO-90 and ML45 benchmark suites. For LIBERO-90, the learner receives 50 expert demonstrations per task from the author-provided dataset. For ML45, we use the scripted policies provided in the official Metaworld codebase to collect 100 demonstrations per task. We evaluate the model at the end of training and for each task run 40 evaluation rollouts (50 for MetaWorld) starting from the initial states selected sequentially from a predefined set. We report the aggregated results across 4 seeds (5 seeds for MetaWorld).

In Figure 2a, we present the average success rate across 90 tasks in LIBERO-90 against the aforementioned baselines. Quantized Skill Transformer achieves state-of-the-art results on LIBERO-90 benchmark, outperforming the baseline VQ-BeT and Diffusion Policy by a margin of 8 and 13% respectively. We attribute its performance to its learned latent space that enables effective knowledge sharing across tasks. While VQ-BeT also shows strong performance, we see that QueST's architecture lends itself better to sharing representations across tasks. Our implementation of ResNet-T achieves significantly better performance than the reported number (16.8%) in [38] but is still lower than QueST. Figure 3a shows the average success rate across 45 tasks in ML45 benchmark. Being a simpler benchmark, all methods perform almost similar in Multitask-IL setting which is consistent with the trend observed in [74].

We attribute the reasonably good performance of the diffusion policy to its nature as a latent variable model, which employs a continuous latent variable with the same dimensionality as the actions. However, the consistent outperformance of QueST over the diffusion policy provides compelling evidence for the benefits of using a bottlenecked latent variable. This bottleneck encourages the model to learn shared representations, resulting in enhanced performance. While both VQ-BeT and PRISE employ a latent bottleneck, VQ-BeT's architecture neglects the inherent inductive biases in the action data, which we believe results in a less well-structured latent space. PRISE incorporates this using a latent forward transition model, but is bottlenecked by the use of BPE which we posit is not suitable for such a dynamic latent space.

## 5.3 Few-shot Transfer to Unseen Tasks

In this setting we take the pretrained model from section 5.2 and test its 5-shot performance on unseen tasks from LIBERO-LONG and held-out set in ML45. We sample only five demonstrations for each task, generate the skill tokens using pretrained encoder and use them to finetune the skill prior and the decoder as described in Section 4.2. We also present the results without finetuning the decoder (frozen $\psi_\theta$ in figure 2b & 3b) to validate its generalization to unseen skill tokens sequences.

Figure 2b shows the average success rate for 5-shot IL across 8 unseen tasks in LIBERO-LONG. QueST achieves SOTA performance, surpassing all other baselines by an absolute margin of 14%.

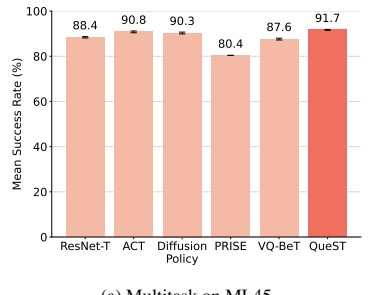

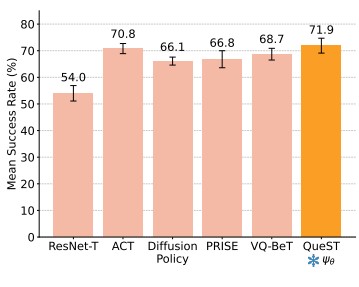

| (a) Multitask on ML45 | (b) Few-shot on ML45 |

Figure 3: Multitask and few-shot success rate on the Metaworld ML45 task suite. In (a) we train on the dataset described in Section 5.2, and in (b) we finetune the model from (a) using 5 demonstrations each from a set of held out tasks. Results show the mean and error bar represents standard error across five random seeds. Results for PRISE are taken from Zheng et al. [74], and the others we reimplemented and ran ourselves.

|  | VQ | Obs. Cond. | Mirror Dec. | Ours |
|---|---|---|---|---|
| LIBERO-90 | $81.2 \pm 0.6$ | $81.9 \pm 1.1$ | $86.3 \pm 0.9$ | $\mathbf{88.6 \pm 0.4}$ |
| Few Shot | $62.5 \pm 2.0$ | $61.3 \pm 2.2$ | $45.4 \pm 2.0$ | $\mathbf{68.8 \pm 1.7}$ |

Table 1: Success rates after ablating design details of QueST. We present the mean across four random seeds and error tolerances show the standard error.

Though we see a marginal drop of $1.5\%$ without decoder finetuning, it still outperforms all the baselines. These results highlight the superiority of QueST in learning transferable representations of action abstractions and effectively leveraging them for downstream decision making. For a fair comparison, we also tried fine-tuning the decoder of VQ-BeT but did not observe any gains from it. VQ-BeT struggles in this setting as it heavily relies on offset head to output continuous action corrections which requires more data-samples for sufficient coverage in the continuous action space. Figure 3b shows the average success rate for 5-shot IL across 5 unseen tasks in MetaWorld. Similar to multitask results, all methods perform comparably, with QueST showing a slight improvement over the others. QueST leverages its learned skill tokens to compositionally model their distribution for an unseen task in just 5 demonstration examples. For few-shot evaluation protocol please refer Appendix D.1.

### 5.4 Long-horizon BC

In this setting, we aim to purely study and compare the performance of our model on long-horizon tasks. We train the model (both stages) solely on LIBERO-LONG complete dataset (50 demonstrations per task) and evaluate with the same scheme as described earlier.

Figure 2c shows the average success rate across 10 LIBERO-LONG tasks. All LVMs perform significantly worse than the ResNet-T model. We attribute this to the relatively small size of LIBERO-LONG dataset which is just not enough to learn a good latent space in LVMs. QueST still outperforms all LVM baselines by a large margin demonstrating its long-horizon modeling capabilities.

Overall, we see that our model outperforms baselines like VQ-BeT in multitask settings, showing stronger modelling capacity. At the same time, it has the correct latent structure to outperform baselines like diffusion in few shot settings, especially even with frozen decoder, indicating strong generalization capabilities of learned skill-space.

### 5.5 Latency

Our pipeline runs at 33Hz, which is more than suffcient for vision-based real-robot control where most camera systems run at 30 fps. For comparison, our implementation of Resnet-T, VQ-BeT, ACT and Diffusion Policy run at 100Hz, 100Hz, 50Hz, and 12Hz respectively.

|  | Non Causal $\phi_\theta$ | Non Causal $\psi_\theta$ | Fully Non Causal | Ours |
|---|---|---|---|---|
| LIBERO-90 | $82.0 \pm 1.6$ | $85.1 \pm 1.8$ | $78.5 \pm 0.5$ | $\mathbf{88.6 \pm 0.4}$ |
| Few Shot | $58.8 \pm 3.0$ | $61.6 \pm 2.5$ | $56.1 \pm 1.8$ | $\mathbf{68.8 \pm 1.7}$ |

Table 2: Success rates after ablating the causality in QueST. We present the mean across four random seeds and error tolerances show the standard error.

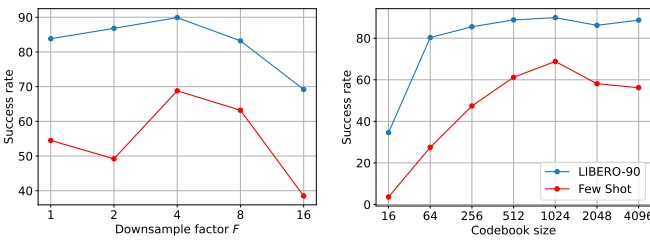

(a) Downsampling factor sensitivity.    (b) Codebook size sensitivity.

Figure 4: We conduct a sensitivity experiment across downsampling factors (a) and codebook sizes (b) on the LIBERO benchmark. For (a) we fix a sequence length of $T = 32$. Overall, we see that the few-shot version is more sensitive to hyperparameters and that $F = 4$ with 1024 codebook vectors are good choices.

## 5.6 Ablations

We validate the proposed architecture by ablating some of its key design decisions. All the ablations are performed on LIBERO studying their effects on both multitask and fewshot IL settings.

1. **Vector Quantization**: We replace the FSQ layer with a Vector Quantization layer of nearly the same codebook size, shown in the *VQ* column of Table 1. We see that FSQ's superior codebook utilization leads to an improvement in performance.

2. **Observation Conditioned Decoder:** Many prior condition the action decoder with current observation [76, 30]. We experiment with this by appending observation tokens to the skill tokens and allowing the transformer decoder to jointly cross-attend to both, shown in Table 1. We see that conditioning on observations leads to a deterioration in performance.

3. **Mirrored Decoder:** Following a typical autoencoder design, we use a decoder that mirrors the encoder, using transposed convolutions instead of strided convolutions, and with the strides in reverse order as in the encoder. This decoder directly takes skill-token embeddings as input and outputs the continuous actions, and results are shown in Table 1. We see that this method performs worse, suggesting attending to all quantized codes in $z$, as our decoder does, is important for faithfully predicting actions.

4. **Causality:** We ablate the use of causal layers in various parts of our network in Table 2. We see that removing causality from any part of our architecture leads to worse performance, suggesting that causal masking imparts the model with a helpful inductive bias for modeling robot action data.

We also perform a sensitivity experiment over several hyperparameters including downsampling factor and codebook size in Figure 4. Across the board we see that the hyperparameters are more important in the difficult few-shot learning setting. In Figure 4a we see that both algorithms have the best performance with a modest downsampling factor of $F = 4$, and in Figure 4b we see that QueST does well with a 1024 codebook vectors. For more discussion on ablations please refer Appendix C.

## 5.7 Latent Skill-Space Analysis

We present a t-SNE visualization (Figure 5) illustrating the learned skill-space across multiple set of similar tasks. We consider four different combinations of similar tasks to effectively examine the z-embeddings corresponding to their trajectories. Each data point in the plot represents a vector of $n$ z-embeddings at a specific timestep throughout the entire episode, with decreasing transparency indicating temporal progression. We show that the QueST encoder learns a semantically meaningful skill-space that encodes shared representations of similar motion primitives across different tasks. This analysis includes the first 11 tasks from LIBERO-90. For better comprehension, we encourage readers to review the corresponding rollouts on the website. Notably, the skill-space learning happens in the first stage training which does not make use of any task labels.

## 6 Conclusion

We present Quantized Skill Transformer, a novel LVM architecture for learning sharable skills in a discrete latent space. The key idea behind QueST is to represent action sequences as a series

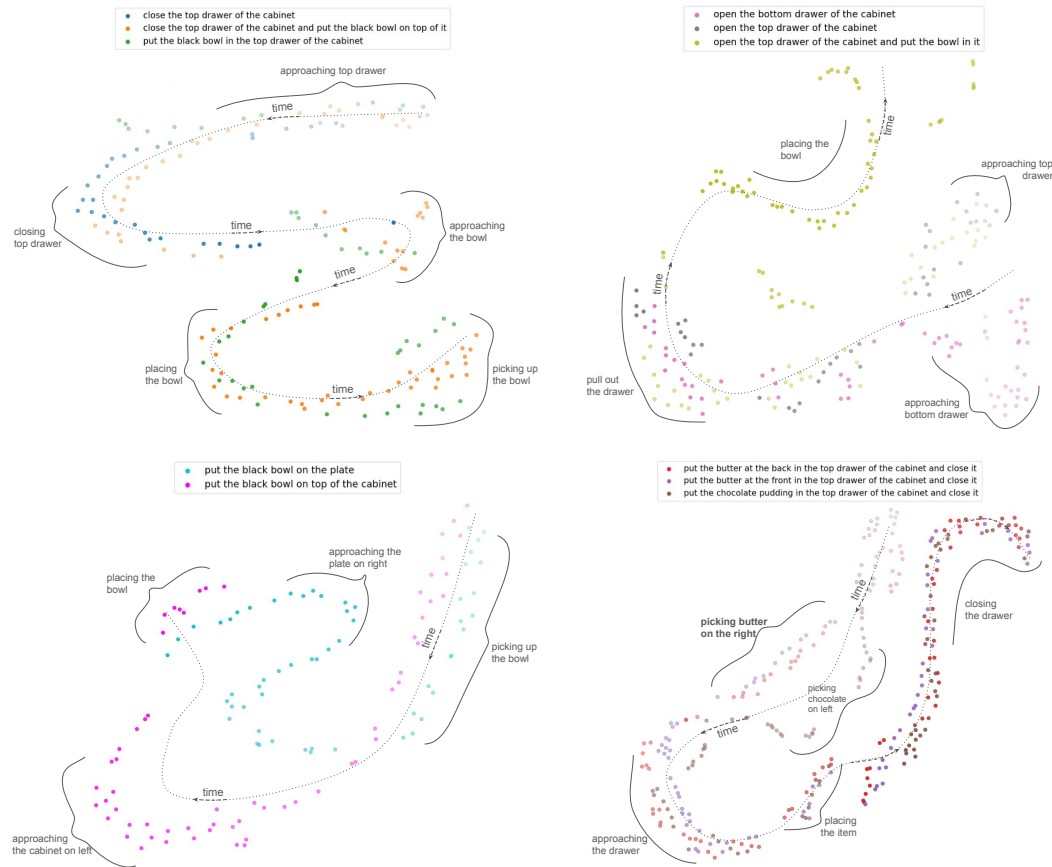

Figure 5: t-SNE visualization of skill-token embeddings. Here, the transparency decreases as the episode progresses. The overall patterns clearly shows how similar motion primitives like approaching, picking and placing from different tasks are aligned with one another.

of codebook vectors, and we demonstrate that using causal convolutions and masked transformers provides an inductive bias that encourages the model to learn useful shared representations. We evaluate QueST across 145 robot manipulation tasks, and show that it outperforms several state-of-the-art baselines in multitask and few-shot learning settings. Our results highlight the usefulness of QueST's encoder (decoder) as semantically-sound, task-agnostic tokenizer (detokenizer) for continuous actions, and its potential to leverage Large Multi-modal Language Models in stage-2.

**Limitations:** While the benchmarks we consider encompass a wide variety of tasks, the held-out tasks are still structurally similar to the pretraining set, which makes few-shot transfer feasible. In scenarios with a more diverse task, current model may struggle to solve new tasks solely within the learned skill space. A promising direction is to train stage-1 on larger datasets, such as Open X-Embodiment [47], with an expanded codebook that could capture more diverse motion primitives. Additionally, our current architecture only accounts for causality. Future work should explore other inductive biases, like geometric invariance and dynamic consistency, to enhance abstraction learning.

**A statement on societal impact:** This paper works towards the broader goal of automating a wide range of manipulation tasks. While this can have positive impacts, such as helping people with mobility impairments or performing menial tasks humans would rather not do, it can also have negative impacts such as automating peoples' jobs away and further concentrating wealth in the hands of a handful of companies. It is important that we in the machine learning community advocate for equitable use of the technology we develop.

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

# Appendix

## A  Website

For further results and videos please see our website. `https://quest-model.github.io`

## B  Experiment Details

### B.1  Hyperparameters:

We present hyperparameters in the following tables:

Table 3: Stage 1 Parameters

| Parameter | Value |
|---|---|
| encoder dim | 256 |
| decoder dim | 256 |
| sequence length ($T$) | 16/32 |
| encoder heads | 4 |
| encoder layers | 2 |
| decoder heads | 4 |
| decoder layers | 4 |
| attention dropout | 0.1 |
| fsq level | [8, 5, 5, 5] |
| conv layers | 3 |
| downsampling factor | 2/4 |

Table 4: Stage 2 Parameters

| Parameter | Value |
|---|---|
| vocab size | 1000 |
| block size ($n$) | 8 |
| number of layers | 6 |
| number of heads | 6 |
| embedding dimension | 384 |
| attention dropout | 0.1 |
| beam size | 5 |
| temperature | 1.0 |
| decoder loss scale | 0/10 |
| execution horizon ($T_a$) | 8 |
| observation history | 1 |

### B.2  Architecture Implementation:

For vision encoder we used a shallow Convolutional Neural Network (CNN), consisting of the first four layers of ResNet18 [27] followed by a spatial softmax [36]. In encoder, we use causal convolution layers from [30]. For transformer blocks, we used the transformers library from hugging face `https://huggingface.co/docs/transformers/` with appropriate masking for ensuring causality.

### B.3  Baseline Implementation:

To ensure fair comparison of different model architectures, we use same input modalities and same observation & task encoders for all baselines, for both LIBERO and Metaworld experiments. VQ-BeT needs a goal image, we instead give it task embedding as goal. Same as QueST, we concatenate observation embeddings for all modalities at any timestep and project them to respective model's hidden dimension.

Depending on the dataset, we also tune some key hyperparameters for the baselines and present the results for best performing ones.

1. **ResNet-T:** Transformer trunk's hidden dimension and number of layers determines the model capacity. Original implementation [38] uses the hidden dimension of $64$ with $4$ layers. We observed improved performance for the hidden dimension of $256$ with $6$ layers and hence report all results for that. As per original implementation we use an observation history of 10 timesteps.

2. **Diffusion Policy:** The model capacity is determined by hidden dimension of U-Net layers. Most widely used implementations use $[256, 512, 1024]$, we ablate a larger model with $[256, 256, 512, 512, 1024]$ but did not observe any performance gains. We also ablate prediction ($T$) and execution horizon ($T_a$) with $16, 32$ and $8, 16$ respectively and observed best performance for $T = 32, T_a = 16$ on LIBERO and $T = 16, T_a = 8$ for MetaWorld. As per original paper ablations [15] an observation history of 1 was used.

3. **VQ-BeT:** Since LIBERO and MetaWorld are larger datasets as compared to the benchmarks in original VQ-BeT paper, we ablate some parameters to increase the model capacity. Specifically, the stage 1 encoder by default is a single MLP layer of dimension $128$. We ablate this with $2, 4$ layers and with $256, 512$ dimensions but observed worse reconstruction loss with increase in capacity. We use residual-VQ configuration of $32/2 \approx 1024$ sized codebook which is close to the codebook size of 1000 for QueST. We use an observation window size of $10$ and ablate the action window size ($T$) with $1, 5, 32$. On LIBERO, the performance was lowest for $T = 1$, and highest for $T = 5$. VQ-BeT maps the whole input sequence to just one embedding leading to extreme compression for larger sequence length and thus performs worse with $T = 32$.

## B.4 Compute:

The models are implemented in PyTorch. For all our experiments we use a server consisting of 8 Nvidia RTX 1080Ti 10GB memory each. And all our models easily fit on one GPU for training.

## B.5 Discussion on Baselines

Many recent works use discrete latent variable models as a mechanism to learn shared abstractions over continuous low-level skills. QueST, VQ-BeT and PRISE all do this and perform two-staged learning. However, in this work we propose several important architecture choices leading to QueST's strong performance. Specifically, QueST encodes actions to a sequence of n encodings (skill tokens) using a novel autoencoder that captures temporal correlation within an action sequence with causal convolution and masked self-attention layers.

Like QueST, VQ-BeT performs temporal abstraction by encoding a sequence of actions into one state-independent latent vector. It concatenates input actions and encodes the sequence to just one single latent encoding using an MLP which does not explicitly model any temporal correlation. A sampled action sequence might contain multiple motion primitives of variable length and start point, and capturing them with a single encoding is limiting as it restricts abstraction at different levels of granularity. This is validated by the poorer ($-14\%$) performance of VQ-BeT with a larger chunk size of 32. QueST flexibly captures this variability within $n$ encodings using its encoder, specifically designed to model temporal correlation within input actions. With an effective codebook size of $C$, QueST's effective latent space is $C^n$ while that of VQ-BeT is $C$. Its relatively small latent space limits its expressive capacity, and thus limits its ability to learn sharable representations between tasks, as evidenced by its worse few-shot performance and reliance on continuous offset prediction (whereas QueST can achieve high success rate using only the output from its action decoder).

PRISE performs temporal abstraction by learning discrete codes for state-action pairs and using BPE to group common token sequences into higher-level skills. However, BPE is known to suffer with evolving language leading to a suboptimal character-level tokenization, and it might struggle to effectively encode new action sub-sequences that the model has not seen before (eg. stitching sequences in between two tasks in LIBERO-LONG). This is primarily why decoder finetuning is necessary in PRISE. On the contrary, QueST is more end-to-end as it lets the encoder handle temporal abstraction in the encoding phase itself and gracefully encodes new action-sequences as combinations of its learned latent codes. Our few-shot results without decoder-finetuning shows that such a unified approach learns more generalized abstractions than the baselines and can more effectively represent new action sequences in unseen tasks.

Despite tuning the hyperparameters, the multitask performance of ACT is very low (54%) on LIBERO. Moreover, as per BAKU [26], ACT's performance drastically reduces when number of demos are reduced to 35-per-task in MetaWorld. This suggests ACT is very data-hungry and hence struggles in low-data regime. Another major issue with ACT is tuning the action chunk size to which the method is very sensitive (-14% in metaworld fewshot with chunk size of 32). While we report QueST results for chunk size 32, we did observe very little variation (<1% in LIBERO) with size 16,48 and 64, indicating robustness to this hyperparameter.

## C Discussion on Ablations

For aiding this discussion we present the ablation results again in table 5 and table 6 below.

|  | VQ | Obs. Cond. | Mirror Dec. | Ours |
|---|---|---|---|---|
| LIBERO-90 | $81.2 \pm 0.6$ | $81.9 \pm 1.1$ | $86.3 \pm 0.9$ | $\mathbf{88.6 \pm 0.4}$ |
| Few Shot | $62.5 \pm 2.0$ | $61.3 \pm 2.2$ | $45.4 \pm 2.0$ | $\mathbf{68.8 \pm 1.7}$ |

Table 5: Success rates after ablating design details of QueST.

- Replacing FSQ with VQ still outperforms VQ-BeT in few-shot setting suggesting that QueST's superior performance is not only due to a better quantization scheme but also due to it architecture that flexibly maps an input sequence to multiple embeddings and allows for efficient transfer.

- It's tempting to ground the mapping between z-tokens and actions with observation tokens with an intuition that z-tokens will define a coarse set of actions and observation tokens will aid finer action decoding. But we observe worse performance with this. We hypothesize that the reconstruction objective forces encoder and decoder for most optimal quantization at the bottleneck layer but with extra observation information the decoder might focus more on observation tokens in turn hurting the quantization. This observation goes hand-in-hand with a closely related prior work SPiRL[58] that tried same ablation and found that state conditioned decoder hurts downstream RL.

- We observe a poorer performance in both multitask and few-shot settings with a conventional stage 1 autoencoder. This validates the QueST's cross-attention architecture that allows for attending to all z-tokens and maintaining causality at the same time.

|  | Non Causal $\phi_\theta$ | Non Causal $\psi_\theta$ | Fully Non Causal | Ours |
|---|---|---|---|---|
| LIBERO-90 | $82.0 \pm 1.6$ | $85.1 \pm 1.8$ | $78.5 \pm 0.5$ | $\mathbf{88.6 \pm 0.4}$ |
| Few Shot | $58.8 \pm 3.0$ | $61.6 \pm 2.5$ | $56.1 \pm 1.8$ | $\mathbf{68.8 \pm 1.7}$ |

Table 6: Success rates after ablating the causality in QueST.

- We observe that a fully-causal stage-1 is most optimal and a non-causal decoder does not hurt as much as a non-causal encoder does. This can be explained with a simplistic setting where the input to stage-1 are 2D trajectories of a point agent. Consider an anti-clockwise circular trajectory and an S-shaped one where the first half of the later overlaps with the first half (semi-circle) of the former. When both of these trajectory sequences are inputted to the stage-1, a non-causal encoder will assign distinct sequences of z-tokens for both trajectories. But a causal encoder will assign same sequence of z-tokens for the first half of both trajectories and distinct to later parts. This allows the model to re-use the z-tokens corresponding to a semi-circle for creating other shaped-trajectories that has semi-circle in them for example C-shaped or infinity-shaped trajectories.

|  | Frozen $\psi_\theta$ | Finetuned $\psi_\theta$ | |
|---|---|---|---|
|  |  | loss scale 10 | loss scale 100 |
| Few Shot | $66.0 \pm 3.6$ | $\mathbf{68.8 \pm 1.7}$ | $66.0 \pm 1.0$ |

Table 7: Success rates for decoder finetuning settings in few-shot IL.

- Table 7 illustrates the impact of decoder finetuning in LIBERO-LONG fewshot IL setting. QueST outperforms all baselines even without finetuning the decoder. Finetuning decoder should not be necessary in this setting, as LIBERO-LONG tasks are combination of two tasks from LIBERO-90 (pretraining set). This highlights QueST's effectiveness in stitching trajectories using its learned skill-space. We report the finetuning results in the main paper, as they exhibit better performance.

# D  Additional Results

## D.1  Fewshot IL

**Fewshot Evaluation Protocol:**  In finetuning phase, we finetune ResNet-T, VQ-BeT & QueST for 100 epochs and ACT & Diffusion Policy for 200 epochs. For each task in MetaWorld, we evaluate

each method across 10 evenly spaced checkpoints for 5 seeds on 50 distinct initial states and report the results corresponding to the best performing checkpoint. For Libero, we found the final checkpoint to perform best for all methods and hence report results corresponding to it across 9 seeds.

Table 8: LIBERO 5-shot IL success rates across unseen 10 tasks. Results across 9 random seeds.

| Task ID | ResNet-T | ACT | Diffusion Policy | PRISE | VQ-BeT | QueST |
|---|---|---|---|---|---|---|
| 1 | $53.8 \pm 11.6$ | $20.0 \pm 6.0$ | $32.0 \pm 5.5$ | $26.7 \pm 6.4$ | $16.5 \pm 11.9$ | $\mathbf{56.4 \pm 5.5}$ |
| 2 | $65.7 \pm 16.9$ | $33.3 \pm 13.1$ | $57.8 \pm 5.9$ | $48.3 \pm 9.4$ | $62.2 \pm 15.8$ | $\mathbf{82.9 \pm 4.7}$ |
| 3 | $70.2 \pm 7.8$ | $67.7 \pm 6.2$ | $\mathbf{76.4 \pm 7.2}$ | $70.0 \pm 0.0$ | $52.7 \pm 7.4$ | $\mathbf{66.7 \pm 6.7}$ |
| 4 | $75.8 \pm 7.6$ | $70.3 \pm 6.2$ | $\mathbf{98.2 \pm 1.7}$ | $78.3 \pm 8.8$ | $45.3 \pm 7.7$ | $88.0 \pm 3.1$ |
| 5 | $26.7 \pm 11.9$ | $35.0 \pm 4.1$ | $\mathbf{44.7 \pm 6.0}$ | $45.0 \pm 10.8$ | $30.3 \pm 13.0$ | $42.0 \pm 7.1$ |
| 6 | $86.9 \pm 4.9$ | $68.3 \pm 6.5$ | $37.1 \pm 3.7$ | $\mathbf{90.0 \pm 4.1}$ | $48.7 \pm 17.2$ | $\mathbf{92.4 \pm 4.4}$ |
| 7 | $24.4 \pm 8.4$ | $15.0 \pm 0.0$ | $14.9 \pm 6.7$ | $25.0 \pm 4.1$ | $45.5 \pm 8.2$ | $\mathbf{58.2 \pm 6.1}$ |
| 8 | $23.7 \pm 12.1$ | $26.7 \pm 7.0$ | $6.2 \pm 3.2$ | $45.0 \pm 8.1$ | $33.3 \pm 9.3$ | $\mathbf{47.1 \pm 8.6}$ |
| 9 | - | - | $\mathbf{55.0 \pm 4.0}$ | - | $15.0 \pm 7.0$ | $46.6 \pm 6.2$ |
| 10 | - | - | $\mathbf{68.3 \pm 6.2}$ | - | $25.0 \pm 0.0$ | $\mathbf{65.0 \pm 12.2}$ |

Table 9: MetaWorld 5-shot IL success rates across 5 unseen tasks. Results across 5 random seeds.

| Task ID | ResNet-T | ACT | Diffusion Policy | PRISE | VQ-BeT | QueST |
|---|---|---|---|---|---|---|
| box-close-v2 | $63.2 \pm 5.2$ | $67.2 \pm 5.2$ | $68.0 \pm 1.6$ | $60.8 \pm 6.6$ | $75.3 \pm 9.6$ | $\mathbf{84.0 \pm 7.3}$ |
| disassemble-v2 | $68.8 \pm 2.0$ | $83.2 \pm 3.2$ | $81.3 \pm 3.8$ | $74.1 \pm 7.3$ | $92.7 \pm 1.9$ | $\mathbf{76.4 \pm 26.0}$ |
| hand-insert-v2 | $37.2 \pm 4.1$ | $53.2 \pm 3.7$ | $39.3 \pm 1.9$ | $\mathbf{60.0 \pm 5.0}$ | $48.0 \pm 6.5$ | $49.6 \pm 6.4$ |
| pick-place-wall-v2 | $42.8 \pm 3.7$ | $74.4 \pm 6.9$ | $70.7 \pm 5.2$ | $71.7 \pm 5.7$ | $65.3 \pm 1.9$ | $\mathbf{76.8 \pm 11.4}$ |
| stick-pull-v2 | $58.0 \pm 8.8$ | $76.0 \pm 3.6$ | $71.3 \pm 1.9$ | $67.5 \pm 5.6$ | $62.0 \pm 11.4$ | $\mathbf{72.8 \pm 11.1}$ |

## D.2 Multitask IL

Table 10: LIBERO-90 multitask IL success rates across 90 tasks. Results across 4 random seeds.

| Task ID | ResNet-T | ACT | Diffusion Policy | PRISE | VQ-BeT | QueST |
|---|---|---|---|---|---|---|
| 1 | $\mathbf{1.00}$ | 0.90 | 0.99 | 0.80 | $\mathbf{1.00}$ | $\mathbf{1.00}$ |
| 2 | 0.96 | 0.30 | $\mathbf{0.98}$ | 0.35 | 0.94 | 0.97 |
| 3 | 0.96 | 0.50 | $\mathbf{0.99}$ | 0.70 | 0.97 | 0.91 |
| 4 | 0.74 | 0.22 | 0.91 | 0.50 | $\mathbf{0.99}$ | 0.94 |
| 5 | 0.95 | 0.58 | 0.93 | 0.45 | 0.95 | $\mathbf{0.98}$ |
| 6 | 0.91 | 0.39 | $\mathbf{0.99}$ | 0.65 | 0.98 | 0.98 |
| 7 | $\mathbf{0.95}$ | 0.29 | 0.94 | 0.50 | 0.86 | 0.93 |
| 8 | 0.96 | 0.72 | 0.90 | $\mathbf{0.95}$ | 0.80 | $\mathbf{0.99}$ |
| 9 | 0.74 | 0.41 | $\mathbf{0.93}$ | 0.60 | 0.75 | $\mathbf{0.93}$ |
| 10 | $\mathbf{0.97}$ | 0.65 | 0.91 | 0.35 | 0.83 | 0.90 |
| 11 | 0.97 | 0.82 | 0.98 | $\mathbf{0.95}$ | 0.96 | 0.97 |
| 12 | 0.90 | 0.73 | 0.94 | $\mathbf{0.95}$ | 0.80 | $\mathbf{0.94}$ |
| 13 | 0.82 | 0.62 | 0.81 | 0.20 | $\mathbf{0.87}$ | 0.76 |
| 14 | 0.86 | 0.72 | $\mathbf{0.94}$ | 0.40 | 0.49 | 0.71 |

| Task ID | ResNet-T | ACT | Diffusion Policy | PRISE | VQ-BeT | QueST |
|---------|----------|-----|------------------|-------|--------|-------|
| 15 | 0.87 | 0.49 | **0.95** | 0.35 | 0.46 | 0.58 |
| 16 | 0.97 | 0.86 | **0.99** | 0.75 | 0.98 | 0.96 |
| 17 | 0.72 | 0.40 | **0.89** | 0.40 | 0.53 | 0.80 |
| 18 | 0.79 | 0.20 | **0.76** | 0.15 | **0.80** | 0.67 |
| 19 | 0.93 | 0.75 | **0.99** | 0.30 | 0.91 | **1.00** |
| 20 | 0.87 | 0.41 | **0.98** | 0.65 | 0.68 | 0.92 |
| 21 | **1.00** | 0.82 | 0.99 | **1.00** | 0.96 | **1.00** |
| 22 | 0.90 | 0.44 | **0.97** | 0.30 | 0.91 | 0.93 |
| 23 | 0.97 | 0.75 | **0.99** | 0.85 | 0.95 | 0.92 |
| 24 | 0.75 | 0.11 | **0.85** | 0.05 | 0.69 | 0.82 |
| 25 | 0.97 | 0.44 | **0.99** | 0.95 | 0.94 | **1.00** |
| 26 | 0.97 | 0.85 | **1.00** | 0.90 | 0.85 | 0.99 |
| 27 | 0.72 | 0.14 | **0.88** | 0.55 | 0.50 | 0.52 |
| 28 | 0.72 | 0.20 | **0.86** | 0.05 | 0.45 | 0.68 |
| 29 | **1.00** | 0.68 | **1.00** | **1.00** | 0.97 | **1.00** |
| 30 | **1.00** | 0.19 | **1.00** | **1.00** | 0.92 | 0.97 |
| 31 | 0.91 | 0.83 | **0.96** | 0.50 | 0.85 | 0.90 |
| 32 | 0.99 | 0.90 | **1.00** | 0.85 | 0.88 | 0.99 |
| 33 | 0.57 | 0.20 | 0.58 | 0.20 | 0.37 | **0.67** |
| 34 | 0.85 | 0.56 | 0.84 | 0.30 | 0.87 | **0.98** |
| 35 | 0.93 | 0.52 | 0.97 | 0.80 | **0.98** | 0.92 |
| 36 | 0.97 | 0.67 | **0.99** | 0.75 | 0.98 | 0.97 |
| 37 | 0.85 | 0.24 | **0.97** | 0.25 | 0.73 | 0.74 |
| 38 | 0.78 | 0.41 | **0.91** | 0.30 | 0.90 | 0.62 |
| 39 | 0.86 | 0.32 | **0.90** | 0.20 | **0.90** | 0.88 |
| 40 | 0.96 | 0.35 | **0.98** | 0.85 | 0.90 | 0.93 |
| 41 | 0.90 | 0.27 | 0.79 | 0.50 | **0.91** | 0.92 |
| 42 | **1.00** | 0.74 | **1.00** | 0.55 | 0.89 | **1.00** |
| 43 | 0.98 | 0.41 | 0.99 | 0.80 | 0.97 | **0.98** |
| 44 | 0.80 | 0.39 | **0.89** | 0.40 | 0.83 | 0.93 |
| 45 | 0.99 | 0.83 | **1.00** | 0.85 | 0.99 | 0.98 |
| 46 | 0.97 | 0.60 | **1.00** | 0.55 | 0.91 | 0.99 |
| 47 | 0.75 | 0.37 | 0.31 | 0.35 | 0.65 | **0.91** |
| 48 | 0.87 | 0.27 | 0.53 | 0.25 | 0.88 | **0.98** |
| 49 | 0.90 | 0.55 | **0.96** | 0.65 | 0.48 | 0.95 |
| 50 | 0.88 | 0.54 | 0.82 | 0.65 | 0.60 | **0.99** |
| 51 | 0.80 | 0.33 | 0.28 | 0.40 | 0.87 | **0.88** |
| 52 | 0.79 | 0.28 | 0.00 | 0.10 | **0.91** | 0.75 |
| 53 | 0.74 | 0.33 | 0.34 | 0.30 | **0.84** | 0.82 |
| 54 | 0.88 | 0.64 | 0.73 | 0.60 | 0.79 | **0.87** |
| 55 | 0.83 | 0.51 | 0.77 | 0.50 | **0.95** | 0.93 |

| Task ID | ResNet-T | ACT | Diffusion Policy | PRISE | VQ-BeT | QueST |
|---|---|---|---|---|---|---|
| 56 | 0.85 | 0.62 | 0.49 | 0.35 | **0.92** | 0.83 |
| 57 | 0.99 | 0.64 | **1.00** | 0.80 | **1.00** | 0.97 |
| 58 | 0.95 | 0.57 | **1.00** | 0.50 | **1.00** | 0.99 |
| 59 | 0.84 | 0.56 | 0.78 | 0.20 | **0.98** | 0.95 |
| 60 | 0.94 | 0.68 | 0.89 | 0.65 | 0.91 | **1.00** |
| 61 | 0.91 | **0.95** | 0.90 | 0.80 | 0.98 | **1.00** |
| 62 | 0.96 | 0.75 | 0.58 | 0.85 | **0.99** | 0.81 |
| 63 | 0.70 | 0.43 | 0.38 | 0.40 | 0.84 | **0.78** |
| 64 | 0.73 | 0.04 | 0.41 | 0.40 | 0.38 | **0.78** |
| 65 | 0.73 | 0.16 | 0.75 | 0.15 | 0.68 | **0.85** |
| 66 | 0.76 | 0.45 | 0.65 | 0.15 | **0.84** | 0.79 |
| 67 | 0.84 | 0.72 | 0.66 | 0.30 | 0.87 | **0.93** |
| 68 | 0.78 | 0.73 | 0.44 | 0.55 | 0.74 | **0.85** |
| 69 | 0.83 | 0.68 | 0.59 | 0.85 | 0.90 | **0.93** |
| 70 | 0.88 | 0.56 | 0.57 | 0.90 | **0.93** | 0.89 |
| 71 | 0.90 | 0.52 | 0.92 | 0.55 | **0.97** | 0.92 |
| 72 | 0.85 | 0.52 | **0.98** | 0.35 | 0.85 | 0.94 |
| 73 | 0.89 | 0.59 | 0.86 | 0.60 | 0.84 | **0.98** |
| 74 | 0.72 | 0.18 | 0.61 | 0.30 | 0.33 | **0.70** |
| 75 | 0.77 | 0.45 | 0.38 | 0.45 | **0.95** | **0.95** |
| 76 | 0.64 | 0.22 | 0.21 | 0.25 | 0.30 | **0.61** |
| 77 | **0.89** | 0.70 | 0.35 | 0.65 | 0.70 | **0.89** |
| 78 | 0.57 | 0.46 | 0.14 | 0.80 | 0.85 | **0.97** |
| 79 | 0.63 | 0.28 | 0.06 | 0.45 | 0.68 | **0.86** |
| 80 | 0.73 | 0.59 | 0.01 | 0.30 | 0.87 | **0.98** |
| 81 | 0.65 | 0.53 | 0.08 | 0.30 | 0.44 | **0.70** |
| 82 | 0.63 | 0.24 | 0.54 | 0.35 | 0.61 | **0.70** |
| 83 | 0.80 | 0.56 | 0.49 | 0.80 | 0.89 | **0.94** |
| 84 | 0.55 | 0.35 | 0.47 | 0.55 | 0.43 | **0.75** |
| 85 | 0.70 | 0.74 | 0.79 | 0.75 | **0.93** | 0.92 |
| 86 | 0.69 | 0.53 | 0.13 | 0.75 | 0.47 | **0.89** |
| 87 | 0.84 | 0.65 | **0.98** | 0.95 | 0.86 | 0.92 |
| 88 | 0.82 | 0.54 | 0.96 | 0.65 | 0.87 | **0.97** |
| 89 | 0.91 | 0.77 | 0.70 | 0.55 | 0.96 | **0.97** |
| 90 | 0.80 | 0.29 | **0.91** | 0.85 | 0.89 | 0.56 |

Table 11: MetaWorld multitask IL success rates across 45 tasks. Results across 5 random seeds.

| Task ID | ResNet-T | ACT | Diffusion Policy | VQBeT | QueST |
|---|---|---|---|---|---|
| assembly-v2 | 0.73 | 0.97 | 0.88 | 0.82 | **1.00** |
| basketball-v2 | 0.76 | 0.80 | 0.78 | **0.82** | 0.68 |

| Task ID | ResNet-T | ACT | Diffusion Policy | VQBeT | QueST |
|---|---|---|---|---|---|
| bin-picking-v2 | 0.89 | **1.00** | 0.96 | 0.20 | 0.94 |
| button-press-topdown-v2 | **1.00** | **1.00** | **1.00** | **1.00** | **1.00** |
| button-press-topdown-wall-v2 | **1.00** | **1.00** | **1.00** | **1.00** | **1.00** |
| button-press-v2 | **1.00** | **1.00** | **1.00** | **1.00** | **1.00** |
| button-press-wall-v2 | **1.00** | **1.00** | 0.98 | 0.98 | 0.98 |
| coffee-button-v2 | **1.00** | **1.00** | **1.00** | **1.00** | **1.00** |
| coffee-pull-v2 | 0.90 | 0.92 | 0.96 | 0.82 | **0.98** |
| coffee-push-v2 | 0.89 | **0.96** | 0.86 | 0.94 | 0.90 |
| dial-turn-v2 | 0.98 | 0.99 | **1.00** | **1.00** | **1.00** |
| door-close-v2 | **1.00** | **1.00** | **1.00** | **1.00** | **1.00** |
| door-lock-v2 | **1.00** | 0.99 | **1.00** | **1.00** | **1.00** |
| door-open-v2 | **0.96** | 0.95 | **0.96** | 0.94 | 0.94 |
| door-unlock-v2 | **1.00** | **1.00** | **1.00** | **1.00** | **1.00** |
| drawer-close-v2 | **1.00** | **1.00** | **1.00** | **1.00** | **1.00** |
| drawer-open-v2 | **1.00** | **1.00** | **1.00** | **1.00** | **1.00** |
| faucet-close-v2 | **1.00** | **1.00** | **1.00** | **1.00** | **1.00** |
| faucet-open-v2 | **1.00** | **1.00** | **1.00** | **1.00** | **1.00** |
| hammer-v2 | 0.95 | **1.00** | 0.98 | **1.00** | 0.94 |
| handle-press-side-v2 | **1.00** | **1.00** | **1.00** | **1.00** | **1.00** |
| handle-press-v2 | **1.00** | **1.00** | **1.00** | **1.00** | **1.00** |
| handle-pull-side-v2 | 0.69 | 0.94 | 0.78 | 0.74 | **0.98** |
| handle-pull-v2 | **1.00** | **1.00** | **1.00** | **1.00** | **1.00** |
| lever-pull-v2 | **0.94** | 0.93 | 0.84 | 0.80 | 0.92 |
| peg-insert-side-v2 | 0.81 | **0.94** | 0.90 | 0.76 | 0.86 |
| peg-unplug-side-v2 | 0.88 | 0.91 | 0.88 | **0.92** | 0.90 |
| pick-out-of-hole-v2 | 0.62 | **0.89** | 0.74 | 0.34 | 0.76 |
| pick-place-v2 | 0.67 | 0.71 | 0.76 | 0.74 | **0.78** |
| plate-slide-back-side-v2 | **1.00** | **1.00** | **1.00** | **1.00** | **1.00** |
| plate-slide-back-v2 | **1.00** | **1.00** | **1.00** | **1.00** | **1.00** |
| plate-slide-side-v2 | 0.98 | **1.00** | 0.98 | 0.98 | **1.00** |
| plate-slide-v2 | **1.00** | **1.00** | **1.00** | **1.00** | **1.00** |
| push-back-v2 | 0.72 | 0.64 | 0.76 | 0.64 | **0.80** |
| push-v2 | 0.84 | 0.90 | 0.84 | 0.76 | **0.92** |
| push-wall-v2 | 0.92 | 0.98 | 0.94 | 0.94 | **1.00** |
| reach-v2 | **0.39** | 0.37 | 0.32 | 0.28 | 0.36 |
| reach-wall-v2 | 0.49 | 0.47 | **0.52** | 0.36 | 0.42 |
| shelf-place-v2 | 0.65 | 0.85 | 0.66 | 0.76 | **0.88** |
| soccer-v2 | 0.42 | 0.25 | 0.42 | 0.36 | **0.52** |
| stick-push-v2 | 0.75 | **1.00** | 0.96 | 0.94 | 0.96 |
| sweep-into-v2 | 0.90 | **0.92** | 0.88 | 0.90 | 0.84 |
| sweep-v2 | 0.98 | **1.00** | 0.98 | **1.00** | **1.00** |

| Task ID | ResNet-T | ACT | Diffusion Policy | VQBeT | QueST |
|---------|----------|-----|------------------|-------|-------|
| window-close-v2 | 1.00 | 1.00 | 1.00 | 1.00 | 1.00 |
| window-open-v2 | 1.00 | 1.00 | 1.00 | 1.00 | 1.00 |

