# OpenReview forum: "QueST: Self-Supervised Skill Abstractions for Learning Continuous Control"
_NeurIPS.cc/2024/Conference — NeurIPS 2024 poster_

### Official Review · Reviewer_utkR · 2024-07-10

**Soundness:** 2
**Presentation:** 3
**Contribution:** 2
**Rating:** 4
**Confidence:** 3

**Summary:**

The paper introduces an approach utilizing latent variable generative models in conjunction with FSQ quantization techniques to learn good representations of action sequences. It also proposes a prior network for autoregressive modeling at the level of behavioral representations. The superiority of the learned representations is validated across three distinct robotic environments. Additionally, the paper presents intriguing observations, such as the impact of causality on representation learning, and supports these findings with ablation studies. This paper is not too different from many previous relevant literature, and the innovation is a little weak.

**Strengths:**

- The overall paper is easy to read, with the description of the methodology being well-written.

- The experimental content of the paper is robust, and the demonstrations clearly indicate a significant performance enhancement of the proposed method over the baseline.

**Weaknesses:**

- The overall pipeline that first learns the abstract representations of behaviors and then models the behavior with the autoregressive model is common. I believe the main difference of this paper contains two points: (1) using FSQ to learn discrete latent space; (2) implement the encoder with a causal transformer instead of a non-causal transformer. So the innovations seem not enough.

- A lot of ablation studies are conducted. However, the related analysis is not sufficient.

**Questions:**

- Why the causality of the encoder transformer is so critical?

**Limitations:**

Yes

---

> ### Author Rebuttal · Authors · 2024-08-07
>
> Thanks much for your time spent reviewing and your thoughtful comments. We’ve uploaded an updated version of the paper to the website in case you’d like to review the changes we mention.
>
> > I believe the main difference of this paper contains two points: (1) using FSQ to learn discrete latent space; (2) implement the encoder with a causal transformer instead of a non-causal transformer. So the innovations seem not enough.
>
> Many recent works use discrete latent variable models as a mechanism to learn shared abstractions over continuous low-level skills. QueST, VQ-BeT and PRISE all do this and perform two-staged learning. However, in this work we propose several important architecture choices leading to QueST’s strong performance. Specifically, QueST encodes actions to a sequence of $n$ encodings (skill tokens) using a novel autoencoder that captures temporal correlation within an action sequence with causal convolution and masked self-attention layers.
>
> Like QueST, VQ-BeT performs temporal abstraction by encoding a sequence of actions into one state-independent latent vector. It encodes these actions to a single output encoding using an MLP which does not explicitly model any temporal correlation, followed by an MLP decoder and continuous offset predictor. Unlike QueST, VQ-BeT’s relatively small latent space limits its expressive capacity, and thus limits its ability to learn sharable representations between tasks, as evidenced by its worse few-shot performance, worse performance with longer action chunk sizes and reliance on continuous offset prediction (whereas QueST can achieve high success rate using only the output from its action decoder).
>
> PRISE performs temporal abstraction by learning discrete codes for state-action pairs and using BPE to group common token sequences into higher-level skills. However, BPE is known to suffer with evolving language leading to a suboptimal character-level tokenization, and it might struggle to effectively encode new actions from unseen tasks. QueST gracefully handles this by encoding new actions as combinations of its learned latent codes, and its superiority in this regard is evidenced by its stronger few-shot performance even without finetuning the decoder, as is necessary for PRISE.
>
> > A lot of ablation studies are conducted. However, the related analysis is not sufficient.
>
> Below we mention detailed analysis which has also been added to the paper.
> - Replacing FSQ with VQ still outperforms VQ-BeT in a few-shot setting suggesting that QueST’s superior performance is not only due to a better quantization scheme but also due to its architecture that flexibly maps an input sequence to multiple embeddings and allows for efficient transfer.
> - It’s tempting to ground the mapping between z-tokens and actions with observation tokens with an intuition that z-tokens will define a coarse set of actions and observation tokens will aid finer action decoding. But we observe worse performance with this. We hypothesize that the reconstruction objective forces encoder and decoder for most optimal quantization at the bottleneck layer but with extra observation information the decoder might focus more on observation tokens in turn hurting the quantization. This observation goes hand-in-hand with a closely related prior work SPiRL[57] that tried the same ablation and found that state conditioned decoder hurts downstream RL.
> - We observe a poorer performance in both multitask and few-shot settings with a conventional stage 1 autoencoder. This validates the QueST’s cross-attention architecture that allows for attending to all z-tokens and maintaining causality at the same time.
> - We discuss the causality ablation in response to the next question.
> - Figure 1b and 1d in the global response PDF illustrates the impact of decoder finetuning in fewshot IL setting. QueST outperforms all baselines in both benchmarks even without finetuning the decoder. Finetuning decoder should not be necessary in LIBERO-LONG setting, as the tasks are combination of two tasks from LIBERO-90 (pretraining set). This highlights QueST’s effectiveness in stitching trajectories using its learned skill-space. In MetaWorld, we use the ML45 benchmark that is specifically designed to evaluate transfer learning capabilities of Meta-RL algorithms, with unseen tasks being only slightly structurally similar to pretraining tasks. QueST outperforms all baselines though by a small margin but with a frozen decoder, this demonstrates the capability of QueST to effectively combine learned tokens for unseen action distributions.
>
> > Why is the causality of the encoder transformer so critical?
>
> We ablate causal masking of encoder and decoder and observe that a fully-causal stage-1 is most optimal and a non-causal decoder does not hurt as much as a non-causal encoder does. This can be explained with a simplistic setting where the input to stage-1 are 2D trajectories of a point agent. Consider an anti-clockwise circular trajectory and an S-shaped one where the first half of the latter overlaps with the first half (semi-circle) of the former. When both of these trajectory sequences are inputted to the stage-1, a non-causal encoder will assign distinct sequences of z-tokens for both trajectories. But a causal encoder will assign the same sequence of z-tokens for the first half of both trajectories and distinct to later parts. This allows the model to re-use the z-tokens corresponding to a semi-circle for creating other shaped-trajectories that have a semi-circle in them, for example C-shaped or infinity-shaped trajectories. Thus causal masking enables better sharing of skill tokens which leads to improved performance.

---

### Official Review · Reviewer_VaLR · 2024-07-12

**Soundness:** 2
**Presentation:** 3
**Contribution:** 2
**Rating:** 6
**Confidence:** 3

**Summary:**

- This work introduces a new method called QueST that uses a latent variable model to learn a set of discrete temporal action abstractions / motion primitives / skills for imitation learning. This is done by training an auto-encoder to encode a sequence of actions using a causality preserving encoder, discretising the embedding using Finite Scalar Quantization (FSQ) to obtain a sequence of discrete latents, and then decoding the latents using a causality preserving decoder to recover the sequence of actions.
- A policy is learned by predicting the latent actions from a history of observations and a language instruction while keeping the latent decoder frozen. An exception to the latter is when fine-tuning a policy in a few-shot setting. At test time, top-k sampling for sampling an optimal sequence of latents.
- The method is validated on the LIBERO and Meta-World environments and compared to several recent approaches: ResNet-T, diffusion policy, ACT, VQ-Bet, and PRISE.

**Strengths:**

Originality:
- This work is a nice combination of existing methods (latent action abstraction, causal encoders/decoders, and Finite Scalar Quantisation).
- The use of causal encoders/decoders for latent abstraction is an interesting idea.

Quality:
- LIBERO and Meta-World are reasonable benchmarks for validating the proposed method.
- The method shows good performance when compared against several recent baselines: diffusion policy, ACT, VQ-BeT, and PRISE.
- The experimental section contains ablations for several key design choices: FSQ vs VQ, conditioning/not conditioning the auto-encoder on observations, mirroring the architecture of the encoder for the decoder, causal/non-causal encoders, and a sensitivity analysis for the auto-encoder downsampling factor and codebook size.
- Results are reported for three random seeds for the experiments on LIBERO.
- Line 541-544: The paper preempts concerns about the Meta-World results due to issues with the scripted expert policies and updated results are provided at the paper website.

Clarity:
- The paper is generally well-written.
- The videos on the project website provide a nice indication of qualitative behaviour.

Significance:
- Considering the good performance and the simplicity of the method, it seems reasonable that the proposed method will be adopted or further developed by the community, in particular if the code is released as stated in the NeurIPS paper checklist.

**Weaknesses:**

Originality:
- Two of the baselines, VQ-BeT and PRISE, also learn temporal action abstractions. It would be nice to have a more detailed and clearer discussion of the differences between these three methods than is currently provided, e.g., differences and similarities could be summarised in a table.

Quality:
- There are no results for PRISE on Multitask LIBERO-LONG.
- The paper website states: “Depending on the dataset, we also tune some key hyperparameters for the baselines”. This is vague and it is unclear why this was not done for every dataset.
- It is not clear whether all the baselines are based on new experiments or whether some of the results are shown as reported in previous works.
- There is no analysis of inference latency of the proposed method and the baselines, which is important for practical applications.
- It might be interesting to have an ablation experiment that uses Residual VQ as used by VQ-BeT instead of FSQ to assess if this choice plays a significant role for the difference in performance.

Clarity:
- Some information about how the hyperparameters of the baselines were tuned are on the paper website, but important information like this should be in the main body of the paper or the appendix.
- The paper website states that “To ensure fair comparison of different model architectures, we use same input modalities and same observation & task encoders for all baselines…”. This is important information that should be in the main body or the appendix of the paper. It is also unclear whether this only applies to the Meta-World experiments or also the LIBERO experiments.
- It is not specified where the “provided scripted policies” for collecting demonstrations on Meta-World as mentioned on the paper website can be found.
- Minor comment: Figure 3b is not described or mentioned in the text.

Significance:
- The proposed method feels a bit like an incremental evolution of VQ-BeT and PRISE. Even though these have only come out very recently, it would still be good to have a clearer positioning with respect to these two works as already mentioned above.

**Questions:**

- Are the updated results on Meta-World going to replace the current results in the main body of the paper?
- Where can the “provided scripted policies” for collecting demonstrations on Meta-World as mentioned on the paper website be found?
- Are all results for the baselines based on new experiments or are any of the results shown as reported in previous works? In the latter case, it would be useful for this to be indicated in the result tables.
- How were the hyperparameters tuned for the proposed method, for the ablations, and for baselines?
- How does the proposed method compare to the baselines in terms of latency?
- Is the code going to be released publicly?

Updates:
- Presentation: 2 -> 3
- Score: 4 -> 6

**Limitations:**

- It would be nice to have a discussion of latency considerations.

---

> ### Author Rebuttal · Authors · 2024-08-07
>
> Thanks much for your time spent reviewing and your thoughtful comments. We’ve uploaded an updated version of the paper to the website in case you’d like to review the changes we mention.
>
> > Two of the baselines, VQ-BeT and PRISE, also learn temporal action abstractions. It would be nice to have a more detailed and clearer discussion of the differences between these three methods
>
> Many recent works use discrete latent variable models as a mechanism to learn shared abstractions over continuous low-level skills. QueST, VQ-BeT and PRISE all do this and perform two-staged learning. However, in this work we propose several important architecture choices leading to QueST’s strong performance. Specifically, QueST encodes actions to a sequence of $n$ encodings (skill tokens) using a novel autoencoder that captures temporal correlation within an action sequence with causal convolution and masked self-attention layers.
>
> Like QueST, VQ-BeT performs temporal abstraction by encoding a sequence of actions into one state-independent latent vector. It encodes these actions to a single output encoding using an MLP which does not explicitly model any temporal correlation, followed by an MLP decoder and continuous offset predictor. Unlike QueST, VQ-BeT’s relatively small latent space limits its expressive capacity, and thus limits its ability to learn sharable representations between tasks, as evidenced by its worse few-shot performance, worse performance with longer action chunk sizes and reliance on continuous offset prediction (whereas QueST can achieve high success rate using only the output from its action decoder).
>
> PRISE performs temporal abstraction by learning discrete codes for state-action pairs and using BPE to group common token sequences into higher-level skills. However, BPE is known to suffer with evolving language leading to a suboptimal character-level tokenization, and it might struggle to effectively encode new actions from unseen tasks. QueST gracefully handles this by encoding new actions as combinations of its learned latent codes, and its superiority in this regard is evidenced by its stronger few-shot performance even without finetuning the decoder, as is necessary for PRISE.
>
> > There are no results for PRISE on Multitask LIBERO-LONG.
>
> We were unable to reproduce the PRISE results when we ran their code so we were only able to include results presented in their paper.
>
> > The paper website states: “Depending on the dataset, we also tune some key hyperparameters for the baselines”. This is vague…
>
> Below we summarize information about hyperparameter tuning we’ve added to the appendix
> - ResNet-T: We observe optimal performance with a transformer trunk with 6 layers and a hidden dimension of 256 and use those parameters for all results. We use an observation history of 10 timesteps.
> - Diffusion Policy: We tuned the U-Net hidden dimension across [256, 256, 512, 512, 1024] but did not observe any performance gains. We also tune prediction and execution horizons with 16, 32 and 8, 16 respectively.
> - VQ-BeT: We tune the encoder MLP dimension across (128, 256, 512) with (2, 4) layers and observe worse reconstruction loss with increase in capacity. We use residual-VQ configuration giving a similar sized codebook to QueST. We use an observation window size of 10 and tune the action window size across (1, 5, 32).
> - ACT: We tune model hidden dimension (256, 512); chunk size (16, 32); and kl weight (10,100).
>
> > It is not clear whether all the baselines are based on new experiments or whether some of the results are shown as reported in previous works.
>
> We were able to successfully implement Diffusion Policy, VQ-BeT, ResNet-T and ACT on both benchmarks. Thus, we ran and reported those results as shown in the global response PDF. We were unable to successfully recreate PRISE results so we report the results from their paper.
>
> > There is no analysis of inference latency of the proposed method and the baselines… How does the proposed method compare to the baselines in terms of latency?
>
> Our pipeline runs at 33Hz, which is more than sufficient for vision-based robot control where most camera systems run at 30 fps. For comparison, our implementation of Resnet-T, VQ-BeT, ACT and Diffusion Policy run at 100Hz, 100Hz, 50Hz, and 12Hz respectively. We’ll add this detail to the paper.
>
> > It might be interesting to have an ablation experiment that uses Residual VQ as used by VQ-BeT instead of FSQ…
>
> We performed an ablation with Vanilla-VQ of the same codebook size as VQ-BeT’s Residual-VQ and observed full codebook usage throughout the training. Hence we don't expect Residual-VQ performance to vary much from the Vanilla-VQ results as Residual-VQ is mainly used to mitigate codebook collapse which doesn’t happen in our case. Moreover, the VQ-BeT paper doesn't mention if they ensured their VQ ablation to be of the same effective codebook size which might have led to poor performance in their ablation experiments. It would still be interesting to validate this hypothesis but due to time constraints it might not be viable by the end of the discussion period.
>
> > The paper website states that “To ensure fair comparison of different model architectures, we use the same input modalities and same observation & task encoders for all baselines…”. This is important information that should be in the…paper. It is also unclear whether this only applies to the Meta-World experiments or also the LIBERO experiments.
>
> We’ve moved this detail to the appendix and will add a line confirming that this is the case for all experiments.
>
> > It is not specified where the “provided scripted policies”…can be found.
>
> They can be found in the official Metaworld codebase.
>
> > Figure 3b is not described or mentioned in the text.
>
> Fixed
>
> > Are the updated results on Meta-World going to replace the current results in…the paper?
>
> Yes
>
> > Is the code going to be released publicly?
>
> This will happen alongside the camera ready version of the paper.

---

> > ### Comment · Reviewer_VaLR · 2024-08-12
> >
> > Thank you for the rebuttal.
> >
> > > “Unlike QueST, VQ-BeT’s relatively small latent space limits its expressive capacity”
> >
> > Have you considered running an experiment to examine how the performance of VQ-BeT changes when increasing the size of the latent space?
> >
> > > “BPE is known to suffer with evolving language leading to a suboptimal character-level tokenization, and it might struggle to effectively encode new actions from unseen tasks”
> >
> > Could you elaborate on to what extent this is an issue? Temporal action abstraction should allow the reuse of existing abstractions for new tasks, i.e., it might not be necessary to encode “new actions”.
> >
> > > “We were unable to successfully recreate PRISE results so we report the results from their paper”
> >
> > Minor comment: I would suggest adding a footnote or similar to indicate that the results shown for PRSE are taken from the original PRISE paper.

---

> > > ### Author Response · Authors · 2024-08-12
> > >
> > > Thank you for your suggestion and further questions.
> > >
> > > > Have you considered running an experiment to examine how the performance of VQ-BeT changes when increasing the size of the latent space?
> > >
> > > There is a slight misunderstanding here. If by increasing latent space you mean trying a larger codebook size then yes, we do try an effective codebook size of 4096 for VQ-BeT but did not observe any performance gain (<1% variation in LIBERO-90). This follows what VQ-BeT authors report for their codebook size ablations (Table 12 in VQ-BeT paper). What we meant by a "smaller latent space" is the fact that VQ-BeT concatenates input actions and encodes the sequence to just one single latent encoding using an MLP. A sampled action sequence might contain multiple motion primitives of variable length and start point, and capturing them with a single encoding is limiting as it restricts abstraction at different levels of granularity. This is validated by the poorer (-14%) performance of VQ-BeT with a larger chunk size of 32. QueST flexibly captures this variability within 'n' encodings using its encoder, specifically designed to model temporal correlation within input actions. With a codebook size of C, QueST’s effective latent space is C^n while that of VQ-BeT is C, hence a smaller latent space. While we report QueST results for chunk size 32, we did observe very little variation (<1% in LIBERO-90) with size 16,48 and 64, indicating robustness to this hyperparameter (a major issue in tuning chunking based methods like ACT, MT-ACT).
> > >
> > > > Could you elaborate on to what extent this is an issue? Temporal action abstraction should allow the reuse of existing abstractions for new tasks, i.e., it might not be necessary to encode “new actions”.
> > >
> > > The reuse of existing abstractions indeed happens, hence PRISE reports a non-zero success rate in the few-shot setting. However, new tasks will definitely contain new action sub-sequences that the model has not seen before (eg. stitching sequences in between two tasks in LIBERO-LONG). BPE relies on frequency statistics from the pretraining data and hence might lead to inefficient tokenization of such new sub-sequences. This is primarily why decoder finetuning is necessary in PRISE. On the contrary, QueST is more end-to-end as it lets the encoder handle temporal abstraction in the encoding phase itself. Our few-shot results without decoder-finetuning shows that such a unified approach learns more generalized abstractions than the baselines and can more effectively represent new action sequences in unseen tasks.
> > >
> > > > Minor comment: I would suggest adding a footnote or similar to indicate that the results shown for PRSE are taken from the original PRISE paper.
> > >
> > > We'll update this in the main paper.
> > >
> > > We hope we have been able to address your questions in the above clarifications. Kindly let us know if you have additional questions or concerns that stand between us and a higher score.

---

### Official Review · Reviewer_jrVG · 2024-07-12

**Soundness:** 2
**Presentation:** 3
**Contribution:** 2
**Rating:** 5
**Confidence:** 3

**Summary:**

The paper aims to capture skill abstractions through training a latent space through encoding and decoding actions. The resulting latent space is used for training a policy that converts observations into the latent space, and uses the trained decoder to output actions. The paper conducts experiments on manipulation suites (LIBERO and MetaWorld) and demonstrates improvement over existing approaches on few-shot and multitask setting.

**Strengths:**

- The idea makes use of the structure of robotic manipulation tasks, where the sequence of actions are often similar for different items and tasks. Notably, the action decoder is not conditioned on the states to learn a latent space that is invariant to the low-level state information.
- The paper is mostly well written

**Weaknesses:**

I am happy to increase my score if these points are addressed.

**Comments**
- Should probably include analysis on what the latent $z$'s ended up learning. Do they actually have some temporal abstraction going on? Can we associate them? It appears that this is in the website but I believe this should be in the main paper as one premise is that the latent representation is leveraging the structure of the actions.
- This work appears to rely on the fact that the demonstrations have some form of structures---probably a limitation. Perhaps in this experimentation setting it is a fair assumption, but what if we have partially observed setting (e.g. items with dynamic inertial properties).
- It is not clear to me how this approach handles multimodality (in the sense of action distributions?)
- Ablation on "causality": I feel the word "causality" is misused because this is simply a mask enforcing whether or not to look at the future data. Is this not still modelling correlation only? How is this causal exactly?

**Questions:**

**Questions**
- Equation 1, perhaps write out what round_std is.
- Page 5, lines 179-180: What is the intuition of using cross attention between positional embedding and skill tokens but not self-attention with positional encoding?
- Page 5, lines 186-187: Do the authors think that this is because the actions can be invariant (e.g. in velocity control the sequence of action is likely similar if we are reaching downwards to the items)? How would stochasticity of the dynamics play into learning this?
- Figure 2: Any intuition why few-shot is better than multi-task for QueST?

**Possible typos**
- Page 8, line 305: viz. means visualized?

**Limitations:**

- The current application is only on robotic manipulation but it will be interesting to extend this to other domains

---

> ### Author Rebuttal · Authors · 2024-08-07
>
> Thanks much for your time spent reviewing and your thoughtful comments. We’ve uploaded an updated version of the paper to the website in case you’d like to review the changes we mention.
>
> > Should probably include analysis on what the latent 𝑧’s ended up learning. Do they actually have some temporal abstraction going on? Can we associate them?
>
> Thanks for the suggestion. First, because the autoencoder learns to compress a sequence of T actions into a smaller sequence of n latent codes (n<T), there is automatically some temporal abstraction. Next, the t-SNE plots on the website clearly show how similar motions are aligned with one another. We visualize an example with two tasks for lifting a bowl and placing it on a plate (on right) and on a drawer (on left) on the website. The z-embeddings align for the motion of approaching and placing the bowl downwards, while deviating for moving leftwards and rightwards, demonstrating how QueST embeds diverse motion primitives into discrete tokens in a semantically meaningful way. Also, we would like to point out that there is no explicit loss for semantic alignment, it’s the reconstruction objective along with the bottleneck layer that guides the learning to extract shared representations. We’ll move these plots and discussion to the appendix.
>
> > This work appears to rely on the fact that the demonstrations have some form of structures-probably a limitation. Perhaps in this experimentation setting it is a fair assumption, but what if we have partially observed setting
>
> While we are making this assumption of structure as an inductive bias for modeling, we do not train the model on any special datasets or use any explicit labels for primitive skills. We train and evaluate on standard datasets that many recent behavior cloning works use and our architecture is designed to capture the structure of these datasets. Motions involved in everyday manipulation tasks naturally have commonalities that can be shared and reused across tasks. This is not an assumption but a property of the data that our model most effectively leverages.
>
> Regarding partial observability, you are correct that we didn't have to deal with partial observability for our experimental settings. The main contribution of our work is a method to tokenize continuous actions. That being said, our method can easily be adapted to handle partial observability. For example, one popular way to do that is to simply pass in a stack of historical frames, and it would be very simple to pass in several observation vectors to our policy prior as opposed to the one we pass in now.
>
> > It is not clear to me how this approach handles multimodality (in the sense of action distributions?)
>
> First, the policy prior outputs latent actions as a categorical distribution and the inherently multimodal categorical distribution over latent actions lends itself well to multimodal data. This phenomenon has been studied in the past in papers such as BeT [56] and VQ-BeT [34]. The Diffusion Policy paper shows how BeT suffers from mode collapse due to lack of temporal consistency which both VQ-BeT and Diffusion Policy resolves by probabilistically predicting a chunk of actions. Likewise, QueST also predicts a categorical distribution over skill-tokens (this chooses mode as per training distribution) and decodes a chunk of actions (this brings temporal consistency). This effect is studied in several prior works [34, 56, 15, 72].
>
> > Ablation on "causality": I feel the word "causality" is misused... Is this not still modeling correlation only? How is this causal exactly?
>
> You are right that causality here means not to look at the future data. There seems to be a namespace collision problem with the word. In the machine learning literature the word causality is used to refer to precisely the type of masking we are using in our architecture. For our convolutions we use the causal convolutions from wavenet [63] and for the transformer we use the square causal mask defined in pytorch, TensorFlow and HuggingFace. However we understand how this might cause confusion so we’ll replace the word ‘causality’ with ‘causal masking’, emphasizing how this boosts performance.
>
> > Equation 1, perhaps write out what round_std is.
>
> It's the nearest integer rounding with straight-through gradients. We’ll update the manuscript.
>
> > Page 5, lines 179-180: What is the intuition of using cross attention between positional embedding and skill tokens but not self-attention with positional encoding?
>
> We follow the original transformer decoder architecture which consists of alternate masked self-attention and cross-attention layers. Thus we self-attend to positional embeddings with a causal mask as attending to future positional embeddings won’t provide any extra information for the i^th embedding.
>
> > lines 186-187: Do the authors think that this is because the actions can be invariant…? How would stochasticity of the dynamics play into learning this?
>
> Yes, state-independent abstractions are generalizable as many distinct tasks share common motion primitives, especially with velocity control which offers further invariances. This is more performant than its state-conditioned counterpart as it forces action information through the bottleneck layer, preventing the decoder from ignoring it and focusing on states, hurting the quantization. A closely related prior work, SPiRL[57], tried the same ablation with similar results.
>
> The stochasticity of dynamics are accounted for at execution time. The model predicts actions for the next 1-2 seconds, executing them in a receding horizon fashion and intermittently replanning to account for stochastic dynamics.
>
> > Figure 2: Any intuition why few-shot is better than multi-task for QueST?
>
> The few-shot version does better because it is also trained on the data from the LIBERO-90 suite, and it transfers action representations to the tasks in the LIBERO-LONG suite. We'll add details to the paper to make this more clear.

---

> > ### Comment · Reviewer_jrVG · 2024-08-12
> >
> > Thank you for the response. In short I have raised my score.
> >
> > Regarding to the analysis: I feel the main paper should include this because that is verifies the claim that the model is able to learn skill abstractions. Even if the "performance" is not as well I believe in that case it would still deliver the point.
> >
> > Regarding structure: Yes, I meant tasks like manipulation/physical tasks have inherit structure, and the method itself is able to leverage this through the data. Would this method still work if the dataset is not stored as trajectories? If I understand correctly this method requires temporal data.

---

> > > ### Author Response · Authors · 2024-08-12
> > >
> > > Thanks for the feedback and for raising your score. Your feedback has been a great help to improve the presentation and clarity of the paper.
> > >
> > > > Regarding to the analysis…
> > >
> > > This is a good point. We’ll move a subset of these plots to the main body of the paper as well as some analysis describing how they demonstrate QueST’s capabilities to learn shared skills across several tasks.
> > >
> > > > Regarding structure…
> > >
> > > Thank you for raising this concern. An example of a dataset without temporal/trajectory data is the ARNOLD[1] benchmark that has current observation as state (s) and next gripper keypoint as action (a) data. While our proposed method does require temporal data, we believe that the same architecture could work for (s,a) pairs. In fact, we initially did test the architecture on ARNOLD, where the input to the encoder was (s,a) pairs and the decoder output was actions (a), and were able to achieve fairly low reconstruction loss. However, we currently leave this application of the architecture to future work. Since we haven’t evaluated QueST rigorously in such a setting, we’ve added a sentence to the Problem Setting section (3.1) emphasizing this assumption.
> > >
> > > Additionally, we’d like to point out that this assumption is extremely common in recent behavior cloning literature. All of our SOTA baselines (VQ-BeT, PRISE, ACT, Diffusion Policy) make a similar assumption, and several recent large scale robotics datasets (Open-X Embodiment, DROID, BridgeData, etc) and popular behavior cloning benchmarks (Robomimic, Mimicgen, LIBERO, Metaworld, CALVIN, RLBench, Franka Kitchen, D4RL, etc.) are compatible with this assumption. Thus, while this assumption may limit the method’s applicability to some settings, it enables scalability and still fits in well with the field as a whole.
> > >
> > > Thanks again for your valuable feedback. We would appreciate hearing about any further limitations we can address in order to further increase the score.
> > >
> > > [1]: Gong, Ran, et al. "ARNOLD: A benchmark for language-grounded task learning with continuous states in realistic 3D scenes." Proceedings of the IEEE/CVF International Conference on Computer Vision. 2023.

---

### Official Review · Reviewer_cTdz · 2024-07-15

**Soundness:** 3
**Presentation:** 3
**Contribution:** 2
**Rating:** 7
**Confidence:** 3

**Summary:**

This work develops a novel framework for learning generalizable skills from demonstration data. The author’s model uses a quantized discrete latent variable model that compresses skills into a sequence of latent variables and predicts temporal sequences of actions. Their approach decodes skill by cross-attending the sequence of latent tokens against fixed positional encodings, which differs from previous works that typically condition states and actions as inputs to the decoder models. In their experimental evaluations on several benchmarks ( LIBERO and MetaWorld), the authors demonstrate their skill learning framework performs better than alternatives, and additional ablation highlights the impacts of the Quantization parameter codebook size thresholds on how much they benefit performance for inference.

**Strengths:**

The paper is well-written and explains the author's framework in precise detail. It was interesting to suggest that a state-less sequence combined with cross-attention could yield such good performance. Given the empirically solid performance and the author's attention to ablating relevant factors of their system, the paper addresses the problems described and justifies the proposed skill model.

**Weaknesses:**

The only major weakness of the author’s work is the limited evaluations. If the authors can justify using just three seeds across experiments, that would build more confidence. Otherwise, as this system targets robotic learning, it would have been good to see results on a real robotic system instead of just in simulation. The authors use large transformer models, so latency concerns could be relevant if their system is computationally slow. We are also concerned with the lack of error bars in Figure 4, which appears only to use a single seed for the ablation experiments, casting doubts on any conclusion the authors make from these results. The authors should clarify these details and run additional experiments to show the robustness of their results if they have not.

**Questions:**

- How many seeds were used in Figure 4 experiments?
- If the latent variables are used as key and query values, is it fair to say the generated skills can be imagined as some interpolation between a subspace of the fixed-sized vectors constructed by the positional encodings?

**Limitations:**

The major limitation we see is the lack of real robot experiments using the author’s system. For real robotic applications, such evaluation is necessary for there to be an acceptance of such methods to be deployed in the real world.

---

> ### Author Rebuttal · Authors · 2024-08-07
>
> Thanks much for your time spent reviewing and your thoughtful comments. We’ve uploaded an updated version of the paper to the website in case you’d like to review the changes we mention.
>
> > The only major weakness of the author’s work is the limited evaluations. If the authors can justify using just three seeds across experiments, that would build more confidence.
>
> For behavior cloning settings, which tend not to show significant variation across random seeds, performing experiments with comparatively low numbers of random seeds is fairly common and accepted in the literature. For example, BAKU and VQ-BeT [34] only use 1 seed per experiment. ACT [72], Diffusion Policy [15] and 3D-Diffusion Policy use 3 seeds while PRISE [73] uses four.
>
> For our multitask experiments, we now report 4 seeds for LIBERO and 5 seeds for MetaWorld. As evidenced by the error bars, these results have very low variance (<1%) across seeds for both benchmarks. Hence, such a small number of seeds are sufficient to convincingly demonstrate the statistical significance of our reported results.
>
> For our fewshot experiments, which have more variance across random seeds, we’ve run further random seeds for LIBERO benchmark. Specifically, we run three fewshot training seeds for each of the three pretraining seeds, resulting in a total of 9 seeds. Please see the updated results in the global response PDF, summary: QueST outperforms best performing baselines by 14% (absolute). Since variance across pretraining seeds is very low (following multitask results) and all stage-2+decoder parameters are finetuned, the fewshot results have very low reliance on pretraining seeds and hence our seeding methodology is justified. For MetaWorld, we report results for both settings across 5 seeds and observe all methods to perform comparatively with QueST slightly better than others.
>
> In order to more rigorously verify these assertions, we’ve performed a t-test, whose results are contained in the PDF response. In summary we see statistically significant improvements in performance across all benchmarks except Metaworld fewshot.
>
> > The latency concerns could be relevant if their system is computationally slow.
>
> Our pipeline runs at 33Hz, which is more than sufficient for vision-based real-robot control where most camera systems run at 30 fps. For comparison, our implementation of Resnet-T, VQ-BeT, ACT and Diffusion Policy run at 100Hz, 100Hz, 50Hz, and 12Hz respectively. We’ll add this detail to the paper under a new subsection in section 5.
>
> > It would have been good to see results on a real robotic system instead of just in simulation.
>
> We agree! Unfortunately we do not have the capacity to run real robot experiments at this time but this is an important limitation we’ll add to the limitations section of the paper.
>
> > Regarding lack of error bars in Figure 4. How many seeds were used in Figure 4 experiments?
>
> Unfortunately this is a very computationally demanding experiment to run and since our university-provided compute is swamped with other authors working on NeurIPS rebuttals, we might not be able to show more seeds for this experiment by the end of the discussion period. That being said, we will add two further random seeds before the camera ready version is released. While this is an unfortunate circumstance, hopefully you'll agree that the lack of seeds in this experiment don't meaningfully detract from our overall claims about the effectiveness of QueST in multitask and fewshot settings.
>
> > If the latent variables are used as key and query values, is it fair to say the generated skills can be imagined as some interpolation between a subspace of the fixed-sized vectors constructed by the positional encodings?
>
> Not quite. Since decoder cross attends to latent skill tokens, their Key vectors (K) and Value vectors (V) are used with Query vectors (Q) from positional encodings. Thus the output actions (skills) lie in the space spanned by the value vectors from skill tokens. Additionally, the softmax and GeLU non-linearity makes the relationship more complex than simple interpolation. A more accurate way to describe might be that the output actions are a weighted combination of transformed vectors derived from skill tokens, where the weights are contextualized based on positional encodings (for temporal correlation).

---

### Author Rebuttal · Authors · 2024-08-07

We are grateful for the insightful feedback from all reviewers. The reviewers have recognized the novelty of our approach in modeling the inherent structure of manipulation action data through temporal correlation and causal-masking. A particularly noteworthy aspect of our research is the learning and decision-making framework based on state-independent abstractions, which has garnered significant interest. Our comprehensive evaluation, spanning 145 manipulation tasks across 3 diverse benchmarks and 3 distinct settings, against 5 established baselines demonstrates the robustness, versatility and superiority of our method. We are encouraged by the reviewers' assessment that QueST's exceptional performance and simplicity make it a promising candidate for wider adoption within the community.

Below we summarize some common concerns:
1. **Innovation/unique contribution**: QueST introduces a novel autoencoder architecture that flexibly captures diverse, variable-length motion primitives by representing them using a sequence of discrete codebook entries (skill tokens). It does so by modeling temporal correlation in action sequence data resulting in shareable and transferable abstractions, leading to superior multitask and fewshot performance.
1. **Latency**: Our pipeline runs at 33Hz, which is more than sufficient for vision-based real-robot control where most camera systems run at 30 fps. For comparison, our implementation of Resnet-T, VQ-BeT, ACT and Diffusion Policy run at 100Hz, 100Hz, 50Hz, and 12Hz respectively.
2. **Latent space analysis**: On our project website, we provide t-SNE visualizations of the skill token embeddings. These visualizations demonstrate clear alignment of similar motion primitives (such as approaching, picking, and placing) across diverse tasks throughout the rollouts. Notably, this coherent skill-space emerges during stage-1 training, without any explicit skill or task labels, or semantic loss functions. The semantic consistency we observe is a direct result of our carefully designed architecture, which inherently promotes and facilitates the sharing of representations across tasks. This emergent structure in the latent space underscores the effectiveness of our approach in flexibley capturing generalized low-level skills without task-specific supervision.
3. **Limited evaluation/statistical significance**: We report LIBERO and MetaWorld multitask results on 4 and 5 seeds respectively. These results show very low variance (as per error bars) demonstrating strong statistical significance. We’ve expanded our fewshot evaluation to further support our findings: the few-shot LIBERO results now encompass 9 seeds, while few-shot MetaWorld utilize 5 seeds throughout. Moreover, we've performed t-tests across all our results to rigorously validate their statistical significance. Please check out the PDF attached with this response for detailed results.

Additionally we would like to point out that QueST performs almost similarly with and without decoder finetuning with both variants still outperforms all baselines in fewshot setting on both benchmarks. This suggests that QueST stage-1 can effectively extrapolate in learned skill space and combine the skill tokens to generalize to unseen action sequences. This highlights the potential for QueST’s encoder (decoder) as an universal tokenizer (detokenizer) for continuous actions once trained on a large enough dataset like OpenX Embodiment. This semantically-sound, task-agnostic, temporally abstracted tokenization can better facilitate the learning of large behavior models like RT2 and OpenVLA as compared to their currently used naive discretization schemes.

---

### Author Response · Authors · 2024-08-11
**Gentle reminder!**

We kindly request all reviewers to go through the rebuttals and let us know if their concerns have been addressed. Please let us know if there are any more questions/clarifications. Thank you.

---

### Decision · Program_Chairs · 2024-09-25

**Decision:**

Accept (poster)

**Comment:**

The authors have diligently addressed many of reviewers' concerns. Experimental details have been clarified, additional analysis provided, and questions regarding novelty and latency resolved. While the initial reviews showed some variance, nearly all reviewers ultimately recommended acceptance, with the authors' detailed rebuttal and supplementary materials playing a key role in raising scores. I entirely agree with this assessment and applaud the authors for their thorough responses and the helpful website (including the t-SNE visualization!). The core strengths of the paper are a clear contribution to this field of research.

Therefore, I recommend accepting this paper.